# Tyrosol blocks *E. coli* anaerobic biofilm formation via YbfA and FNR to increase antibiotic susceptibility

Ha-Young Choi ⬤[1] & Won-Gon Kim ⬤[1] ✉

Bacteria within mature biofilms are highly resistant to antibiotics than planktonic cells. Oxygen limitation contributes to antibiotic resistance in mature biofilms. Nitric oxide (NO) induces biofilm dispersal; however, low NO levels stimulate biofilm formation, an underexplored process. Here, we introduce a mechanism of anaerobic biofilm formation by investigating the antibiofilm activity of tyrosol, a component in wine. Tyrosol inhibits *E. coli* and *Pseudomonas aeruginosa* biofilm formation by enhancing NO production. YbfA is identified as a target of tyrosol and its downstream targets are sequentially determined. YbfA activates YfeR, which then suppresses the anaerobic regulator FNR. This suppression leads to decreased NO production, elevated bis-(3′−5′)-cyclic dimeric GMP levels, and finally stimulates anaerobic biofilm formation in the mature stage. Blocking YbfA with tyrosol treatment renders biofilm cells as susceptible to antibiotics as planktonic cells. Thus, this study presents YbfA as a promising antibiofilm target to address antibiotic resistance posed by biofilm-forming bacteria, with tyrosol acting as an inhibitor.

Nitric oxide (NO), a well-known signaling molecule in both prokaryotes and eukaryotes, can regulate biofilm dynamics in a wide variety of bacteria, including *Escherichia coli* and *Pseudomonas aeruginosa*[1–3]. Exogenous addition of NO at nontoxic concentrations (approximately nanomolar to micromolar) stimulates biofilm dispersal in *P. aeruginosa*[4]. Specifically, exogenous and endogenous NO activates phosphodiesterase (PDE) activity by directly binding a NO sensor domain of PDE and subsequently decreases cyclic-di-GMP levels, leading to motility and biofilm dispersal in *P. aeruginosa*[2,4–6]. Accordingly, *P. aeruginosa* biofilm dispersal requires the expression of nitrite reductase (NIR), an NO-generating enzyme in bacteria[4,5]. Besides its role in biofilm dispersal, NO is also significant in biofilm formation[7,8]. For *P. aeruginosa*, NO generated by NIR is essential for biofilm formation in both anaerobic and aerobic conditions, presenting a paradox where NO contributes to both formation and dispersal[7,8].

While the mechanism of biofilm dispersal through NIR-derived NO is relatively well understood[1,9], the role of endogenous NO in biofilm formation remains largely unexplored. Recent research has introduced a novel perspective on biofilm formation driven by low

levels of NIR-derived NO[10]. This mechanism was revealed through the investigation of the inhibitory effect of complestatin on biofilm formation. Specifically, the study found that the nitrite transporter NirC, targeted by complestatin, promotes biofilm formation by reducing NO levels[10]. This reduction arises from the partial suppression of *nirB* expression, followed by the activation of diguanylate cyclase (DGC) activity. Subsequently, this triggers elevated cellular c-di-GMP levels in both *E. coli* and *P. aeruginosa*[10]. While this new insight has shed light on the role of NO in biofilm formation, further exploration is needed to understand how other uncharacterized proteins like YhfS, YbfA, PrfC, Wzb, and ZapE influence biofilm formation by repressing NO production in *E. coli*[10]. The mechanisms that lead to the reduction of NO production for biofilm formation remain an intriguing area yet to be fully investigated.

Biofilm formation is regulated by an intercellular chemical communication system quorum sensing (QS) or the intracellular second messenger c-di-GMP. Within biofilms, bacteria exhibit distinctive traits compared to planktonic cells, particularly in terms of growth and gene expression[11]. QS systems regulate the expression of genes related to

[1]Infectious Disease Research Center, Korea Research Institute of Bioscience and Biotechnology, Yusong, Daejeon 34141, Republic of Korea.
✉e-mail: wgkim@kribb.re.kr

the biofilm matrix, shaping its architecture[12]. The pivotal role of c-di-GMP becomes apparent in governing biofilm formation and dispersal[13]. High levels of c-di-GMP levels prompt the shift from planktonic to biofilm mode by downregulating the expression of motility-associated genes and upregulating the expression of exopolysaccharide- and biofilm maturation-associated genes[14]. Conversely, low levels of reduced c-di-GMP levels induce biofilm dispersal through activation of motility mechanisms like flagella and pili[15]. DGCs synthesize c-di-GMP, while PDEs break it down[16]. Reportedly, DGCs and PDEs can be regulated by the QS intercellular chemical communication system[17] or activated by the signaling molecule NO[2,5,10].

$E.\ coli$ and $P.\ aeruginosa$ produce NO via the denitrification pathways, in which nitrate ($NO_3^-$) is reduced to dinitrogen ($N_2$) via nitrite, NO, and nitrous oxide ($N_2O$) in four reactions. These individual steps are catalyzed by nitrate reductase (NAR), NIR, nitric oxide reductase (NOR), and nitrous oxide synthase (NOS)[18]. In $E.\ coli$, fumarate and nitrate reductase regulator (FNR) primarily regulates this process[19]. In contrast, in $P.\ aeruginosa$, it's regulated by arginine nitrate regulator (ANR) and dissimilative nitrate respiration regulator (DNR)[2,20]. Under anaerobic conditions, such as those encountered in the CF airway mucus and inside biofilms, $P.\ aeruginosa$ can obtain sufficient energy through the denitrification pathway using nitrate ($NO_3^-$)/nitrite ($NO_2^-$) as final electron acceptors[21,22]. The denitrification pathway is also active under aerobic conditions[2].

Biofilms cells display remarkable resistance to antibiotics and host antimicrobial defense mechanisms to which planktonic cells are susceptible, often leading to persistent and chronic infections that are difficult to eradicate[23–25]. While multiple factors, such as reduced metabolic rates, slowed growth due to nutrient starvation, the presence of persister cells, and restricted penetration of antimicrobials into the biofilm, contribute to the high level of antimicrobial tolerance of biofilms, the developmental stages of biofilm formation also affects biofilm antimicrobial tolerance[26,27]. The biofilm developmental stages are referred to as reversible and irreversible attachment, biofilm maturation-I and II involving cluster and microcolony formation, respectively, and dispersion[26]. Following reversible attachment, there is a reduction in flagella gene expression and production of biofilm matrix components, leading to the irreversible attachment stage during which drug tolerance is induced through the expression of multidrug efflux pump genes[27]. Mature biofilms are also highly resistant to antimicrobials due to oxygen limitation in their interior[28–30]. Oxygen depletion reportedly accounts for at least 70% of the antibiotic resistance observed in mature biofilm cells of $P.\ aeruginosa$[31]. $P.\ aeruginosa$ is responsible for biofilm-associated chronic infections, including cystic fibrosis (CF) lung infections where it stands as a leading cause of death[32,33]. $E.\ coli$ causes challenging biofilm-based, medical device-related urinary tract and intestinal infections[34,35]. The inclusion of multidrug-resistant $P.\ aeruginosa$ and $Enterobacteriaceae$ in the WHO's antibiotic-resistant bacteria priority list highlights the urgency for strategies against biofilm-related infections[36,37].

Here, we show a mechanism regulating NO production in biofilm formation by exploring the antibiofilm mechanism of tyrosol[38], which is commonly found in foods such as olive oil and wine. Using an $E.\ coli$ mutant library, we demonstrate that the poorly characterized protein YbfA is a tyrosol target. Further studies reveal that YbfA stimulates biofilm formation by maintaining low NO production via YfeR and FNR. Inspired by the involvement of the anaerobic transcriptional regulator FNR, we find that the YbfA-mediated biofilm formation is activated during the late stage of biofilm development, where biofilms are formed under anaerobic conditions. Importantly, this study reveals that tyrosol increases the sensitivity of biofilm cells to antibiotics and highlights YbfA as a promising antibiofilm target for addressing multidrug-resistant biofilms.

## Results

### Tyrosol potently inhibits $E.\ coli$ and $P.\ aeruginosa$ biofilm formation by enhancing endogenous NO production

To discover fungal fermentation extracts that stimulate NO production, thereby reducing biofilm production by $P.\ aeruginosa$ PA14, we screened 2,056 extracts. This process led us to select extracts from two specific fungal strains. However, only one strain's fermentation extract inhibited biofilm formation by prompting NO production, without affecting cell viability. Bioactivity-guided fractionation of a fermentation culture of this strain, identified as $Cladosporium$ sp. FN329, revealed tyrosol (Fig. 1a): tyrosol potently reduced $E.\ coli$ biofilm formation in a concentration-dependent manner by 13.0, 28.3, and 48.9% at 0.1, 1, and 10 μM, respectively, whereas planktonic cell growth was not affected (Fig. 1b). This was tenfold more effective than furanone C-30 (FC), a well-known QS inhibitor. Tyrosol also exhibited similar potency against PA14 (Fig. S1). Its biofilm-inhibiting effect wasn't due to a reduction in total cell number or cell viability, as confirmed by optical density and viable cell assays (Fig. S1a). Notably, the potent antibiofilm activity of tyrosol in both $E.\ coli$ and PA14 was dramatically abrogated by the addition of 2-(4-carboxyphenyl)−4,4,5,5-tetramethylimidazoline-1-oxyl-3-oxide (C-PTIO), a known NO scavenger (Fig. 1b and S1bi). These results suggest that tyrosol inhibits biofilm formation by inducing the production of endogenous NO.

The effects of tyrosol on biofilm formation and the involvement of NO in its inhibition were verified using confocal laser scanning microscopy (CLSM). $E.\ coli$ biofilms were formed on glass coverslips within 24-well plates and then stained with SYTO 9/propidium iodide to assess biofilm architecture. After statically growing $E.\ coli$ cells for 24 h, a biofilm with a thickness of 26.48 μm developed. A dose-dependent reduction in biofilm depth was observed by tyrosol treatment, showing a 65.7% inhibition at 10 μM compared to untreated biofilms (Fig. 1ci and cii), similar to the inhibition pattern observed in the crystal violet assay in the 96-well plates. No red fluorescence was observed, indicating the absence of dead cells. In parallel, the fluorescent NO probe 4,5-diaminofluorescein diacetate (DAF-2DA) was used to measure intrabiofilm NO levels. We observed a dose-dependent increase in NO levels in $E.\ coli$ cells treated with tyrosol compared to untreated cells (Fig. 1ci and cii). Similarly, when we used sodium nitroprusside (SNP) at 5 μM as an NO donor, serving as a positive control, it resulted in elevated NO levels and subsequent inhibition of biofilm formation. In contrast, treatment with FC did not affect NO production but inhibited biofilm formation as expected.

To further confirm the NO-mediated inhibitory activity of tyrosol on biofilm formation, we analyzed the components of the extracellular polymeric substances (EPS) in $E.\ coli$ and PA14 biofilms. The amounts of extracellular carbohydrates, proteins, and eDNA in both free and bound EPS in $E.\ coli$ and PA14 biofilms were significantly decreased by tyrosol treatment (0.1–10 μM) compared with those in an untreated control, and the effects were reversed by C-PTIO treatment (Fig. S2). Taken together, these results indicate that tyrosol inhibits $E.\ coli$ and $P.\ aeruginosa$ biofilm formation by inducing the production of endogenous NO without inhibiting cell growth.

### Tyrosol increases PDE activity and subsequently decreases c-di-GMP levels by increasing NO production

NO reportedly inhibits biofilm formation by enhancing the activity of the c-di-GMP-degrading enzyme PDE and subsequently reducing cellular c-di-GMP levels (Fig. 1d)[2,5]. DGCs and PDEs are responsible for c-di-GMP biosynthesis and degradation, respectively[16,39]. Thus, to understand how the effect of tyrosol on endogenous NO production results in decreased biofilm formation, the effects of tyrosol on cellular c-di-GMP levels and PDE and DGC activity were investigated in the absence or presence of C-PTIO. Tyrosol (0.1-10 μM) lowered c-di-GMP levels and raised PDE activity in $E.\ coli$ and PA14 biofilms, without affecting DGC activity (Figs. 1ei−iii and S1bii−iv). Additionally, consistent with

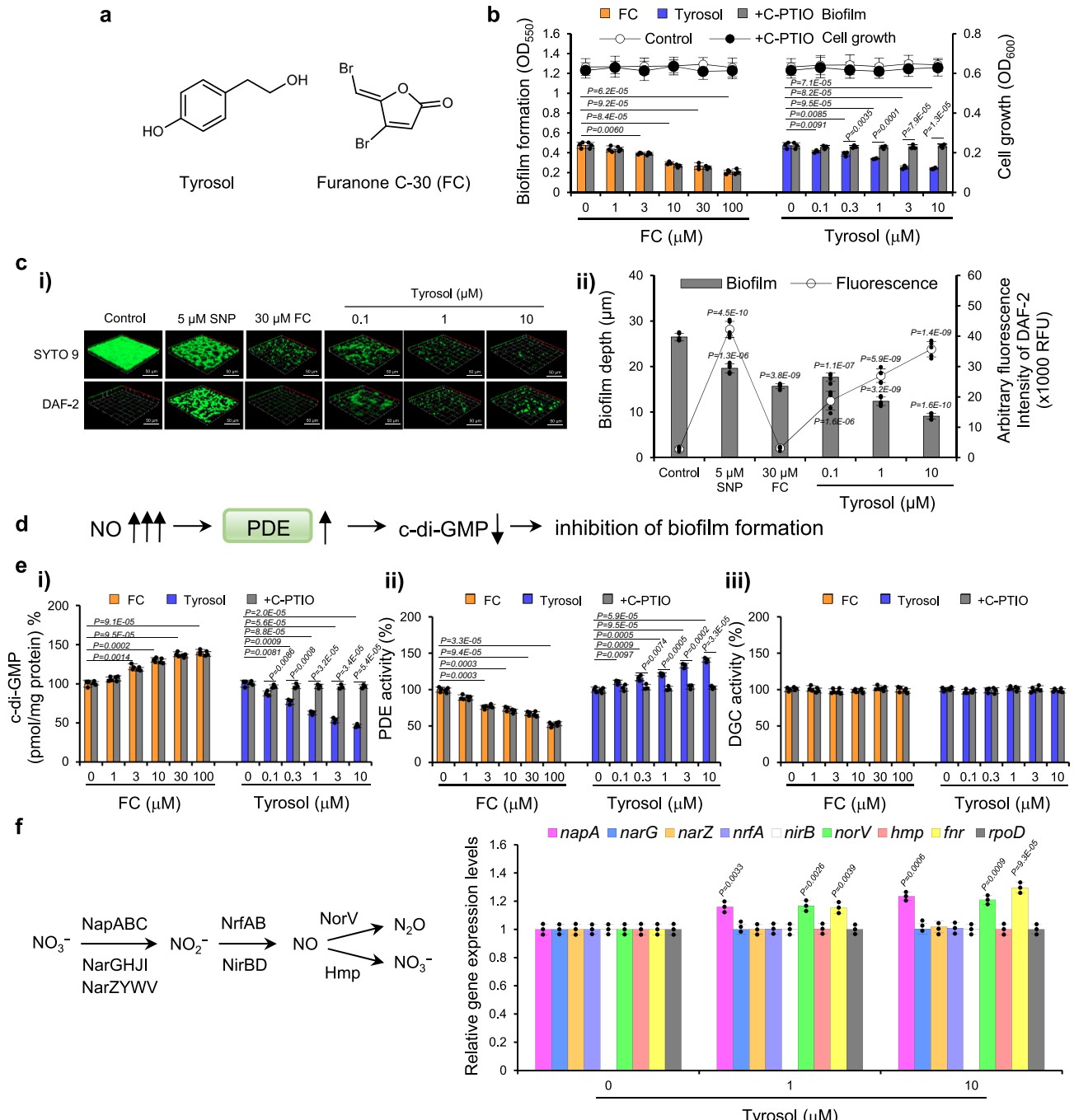

**Fig. 1 | Tyrosol inhibits *E. coli* biofilm formation by lowering cellular c-di-GMP levels via stimulation of PDE activity, which is blocked by the NO scavenger C-PTIO. a** Chemical structures of tyrosol and furanone C-30 (FC), a well-known QS inhibitor. **b** Biofilm formation in *E. coli* BW25113 formed in the presence of tyrosol or FC (control) and in the absence or presence of C-PTIO (2-(4-carboxyphenyl) −4,4,5,5-tetramethylimidazoline-1-oxyl-3-oxide). **c** Confocal laser scanning microscope analyzes of *E. coli* biofilms treated with tyrosol show biofilm inhibition and NO (nitric oxide) production. FC and SNP (sodium nitroprusside), an NO donor, are served as positive controls. **i** Tyrosol-treated *E. coli* biofilms were stained with either SYTO9 (green fluorescence with viable cells, upper row) or DAF-2 (green fluorescence to detect intracellular NO levels, lower row). DAF-2 indicated 4,5-diaminofluorescein diacetate. The experiments were performed twice, and representative images are shown. The scale bar represents 50 μm. **ii** Biofilm thickness and quantification of DAF-2 green fluorescence. Data represent the averages derived from image stacks, collected from five randomly selected areas in each of three

independent biological replicates. **d** NO-mediated inhibition of biofilm formation. Excess NO inhibits biofilm formation via enhancement of PDE activity and subsequent decreases in cellular c-di-GMP levels[2,5]. **e** C-di-GMP levels **i**, and PDE **ii** and DGC **iii** activities in *E. coli* BW25113 formed in the presence of tyrosol or FC (control) and in the absence or presence of C-PTIO. **f** Effects of tyrosol on the expression of denitrification-associated genes and related genes in *E. coli* biofilms formed in the presence of tyrosol for 18 h, as assessed by RT–qPCR. In **b**, **e**, and **f**, the experiment shown is representative of three independent experiments, and data are presented as the mean ± SD of three independent biological replicates. *P*-values were determined using a two-sided Student's *t*-test and, unless otherwise indicated, data were compared with untreated cells. In **cii**, the experiment shown is representative of two independent experiments, with mean ± SD values displayed in each bar. *P*-values were determined using a two-sided Student's t-test and data were compared with control. Source data are provided as a Source Data file.

the reversal of tyrosol-mediated biofilm inhibition by C-PTIO, C-PTIO reversed these changes by tyrosol, unlike the control FC.

It is important to note that specific functions of c-di-GMP are controlled by particular DGCs and/or PDEs[40]. Among the numerous DGCs and PDEs found in *E. coli* and *P. aeruginosa*, only a few are implicated in biofilm formation and dispersion[15,41]. Therefore, to test whether the global reduction in c-di-GMP levels by tyrosol contributes to its antibiofilm activity, the effects of tyrosol on the biofilm formation of a c-di-GMP-overproducing mutant (Δ*wspF*)[42] were investigated. Tyrosol did not inhibit biofilm formation in Δ*wspF* at concentrations up to 10 μM, unlike in the PA14 wild-type strain (Fig. S3ai). Consistently, tyrosol did not reduce the intracellular c-di-GMP levels in the Δ*wspF* strain (Fig. S3bi). In contrast, FC exhibited nearly identical effects on biofilm formation and c-di-GMP levels in both Δ*wspF* and the wild-type strain, as expected, despite its impact on PDE activity (Fig. S3aii and bii). This indicates that tyrosol inhibits biofilm formation by lowering total c-di-GMP levels, while FC affects c-di-GMP levels independently of biofilm formation. Consistent with the global reduction in cellular c-di-GMP levels caused by tyrosol, global PDE activity was enhanced by tyrosol in *E. coli* and PA14 biofilms (Figs. 1eii and S1aiii). Taken together, these results indicate that tyrosol reduces c-di-GMP levels by enhancing the activity of PDEs that are involved in biofilm formation, whereas FC affects PDEs that are not associated in biofilm formation.

Overall, these results indicate that tyrosol inhibits biofilm formation in *E. coli* and PA14 by enhancing NO production, increasing PDE activity, and reducing c-di-GMP levels.

## The nitrate reductase NapA contributes to NO production, which mediates tyrosol-mediated biofilm inhibition

After determining that tyrosol induces NO production, we next sought to examine the involvement of individual steps in the denitrification pathway in *E. coli*. To determine whether tyrosol induces NO production via the denitrification pathway in *E. coli*, we analyzed the mRNA levels of the genes encoding enzymes in the denitrification pathway in tyrosol-treated *E. coli* by real-time quantitative PCR (RT–qPCR). In *E. coli*, there are three NARs (two cytoplasmic NARs (encoded by *narGHJI* and *narZYWV*) and a periplasmic NAR (encoded by *napABC*)) and two NIRs (a cytoplasmic NADH-dependent NIR (encoded by *nirBD*) and a periplasmic cytochrome c-dependent NIR (encoded by *nrfABCDEFG*)[19,43]. NO-detoxifying proteins, such as a NO reductase NOR (encoded by *norVW*) and the flavohemoglobin Hmp, eliminate toxic NO in the denitrification pathway to support anaerobic growth in *E. coli*[44,45]. We analyzed the mRNA levels of the first gene in each operon. Tyrosol at 1 and 10 μM elevated the transcription of a periplasmic NAR (*napA*) and a NOR (*norV*), along with the global anaerobic transcriptional regulator FNR, compared to untreated cells in *E. coli* biofilms (Fig. 1f). Similarly, tyrosol enhanced the transcription of *napA* and the NO reductase-encoding gene *norB* together with that of DNR and ANR, a homolog of *E. coli* FNR, at 1 and 10 μM in a dose-dependent manner in PA14 biofilms (Fig. S1c). These results indicate that tyrosol stimulates the transcription of *napA* and *norV* in the denitrification pathway in *E. coli* and PA14 biofilms, a distinct effect from complestatin's stimulation of *nirB* in planktonic cells, not biofilms, for biofilm inhibition[10].

NapA produces nitrite, a substrate of NO, from $NO_3^-$, while NorV scavenges excess NO[44]. To determine whether enzyme is responsible for tyrosol-mediated NO production and biofilm formation inhibition, the effects of tyrosol on biofilm formation in *napA* and *norV* mutants were tested. Tyrosol did not exhibit antibiofilm activity in the *napA* mutant, whereas FC did (Fig. 2a). In contrast, tyrosol still inhibited biofilm formation in the *norV* mutant (Fig. 2b). These results suggest that NapA is crucial for tyrosol-mediated inhibition of biofilm formation.

This result was confirmed using a *napA*-complemented mutant and a *napA*-overexpressing strain. The elevation in *napA* mRNA levels

induced by tyrosol implied that tyrosol affects *napA* transcription. It has also been reported that the denitrification-associated genes *nap*, *nirB*, and *nrf* are regulated by FNR, whose binding sites are located within -132.5 bp of target genes in *E. coli*[46,47]. Thus, a *napA* construct containing the -150 bp upstream region as a promoter region was used to construct complementation or overexpression strains using the plasmid pBAD with the arabinose-inducible promoter. Indeed, when only the coding component of the *napA* gene was complemented in the *napA* mutant in *E. coli* BW25113, the presence or absence of arabinose at any dose tested in the *napA* mutant did not modulate the effects of tyrosol treatment (Fig. S4a). In contrast, tyrosol antibiofilm activity was rescued in the *napA* mutant complemented with *napA* containing the promoter region (p*napA*) regardless of arabinose addition (Fig. S4b) and, importantly, was reversed by C-PTIO treatment (Fig. 2ai). Furthermore, tyrosol did not inhibit biofilm formation in the p*napA*-overexpressing *E. coli* strain, whereas the control FC did (Fig. 2a). In contrast, tyrosol still showed antibiofilm activity against the p*norV*-overexpressing *E. coli* strain as expected (Fig. 2b). These findings collectively demonstrated that tyrosol requires *napA* to inhibit biofilm formation. Similarly, the dependence on *napA* of inhibition of c-di-GMP levels and elevation of PDE activity induced by tyrosol was demonstrated using the *napA* mutant, the *napA* mutant complemented with p*napA*, and a p*napA*-overexpressing strain (Fig. S5).

This result is consistent with the effects of tyrosol on *napA* expression observed in the RT–qPCR experiments. The *napA* mRNA level was undetectable in the *napA* mutant but restored to almost normal upon complementation with p*napA*, as expected (Fig. 2c). Indeed, *napA* expression in the *napA* mutant complemented with *napA* containing a promoter region (Δ*napA*/pBAD-p*napA*) was increased by tyrosol treatment at 10 μM (Fig. 2c), which is consistent with rescue of the biofilm inhibitory activity of tyrosol in the *napA* mutant complemented with p*napA*. In contrast, in the *napA* mutant complemented with *napA* without a promoter (Δ*napA*/pBAD-*napA*), *napA* expression was not affected by tyrosol treatment (Fig. S4c). These results demonstrate that tyrosol induces biofilm inhibition by stimulating *napA* transcription.

Overall, these data indicate that tyrosol inhibits biofilm formation by activating *napA* expression and promoting the subsequent production of NO, which are followed by increases in PDE activity and subsequent decreases in c-di-GMP levels.

## Tyrosol inhibits biofilm formation by targeting YbfA in E. coli

Next, to further explore how tyrosol inhibits biofilm formation, we investigated potential target proteins of tyrosol using the *E. coli* Keio mutant library, as reported previously[10]. The six *E. coli* mutants (Δ*yhfS*, Δ*nirC*, Δ*ybfA*, Δ*prfC*, Δ*wzb*, and Δ*zapE*) that were previously identified by screening the Keio mutant library showed the same phenotypes as tyrosol-treated cells; 30% less biofilm formation than that of the wild-type BW25113 strain, cell growth that was unaffected by tyrosol treatment, and restoration of biofilm formation to wild-type levels upon C-PTIO treatment[10]. This result suggests that any of the six target proteins might be a target of tyrosol. Thus, to identify an antibiofilm target of tyrosol, the effects of tyrosol on the six *E. coli* mutants were tested to determine whether the mutants are unaffected by tyrosol exposure. Importantly, biofilm formation of only the *ybfA* mutant was unaffected by tyrosol treatment, whereas that of the other mutants was further reduced; notably, that of the *ybfA* mutant was inhibited by FC (Fig. S6a). Additionally, c-di-GMP levels and PDE activity in the tyrosol-treated *ybfA* mutant were not changed compared to those in untreated cells (Fig. S6b). These results suggest that tyrosol can inhibit biofilm formation by targeting YbfA.

This hypothesis was confirmed using six overexpression strains in which the genes mutated in these six mutants (Δ*yhfS*, Δ*nirC*, Δ*ybfA*, Δ*prfC*, Δ*wzb*, and Δ*zapE*) were separately overexpressed in *E. coli* BW25113 using the plasmid pBAD. Indeed, in the presence of arabinose,

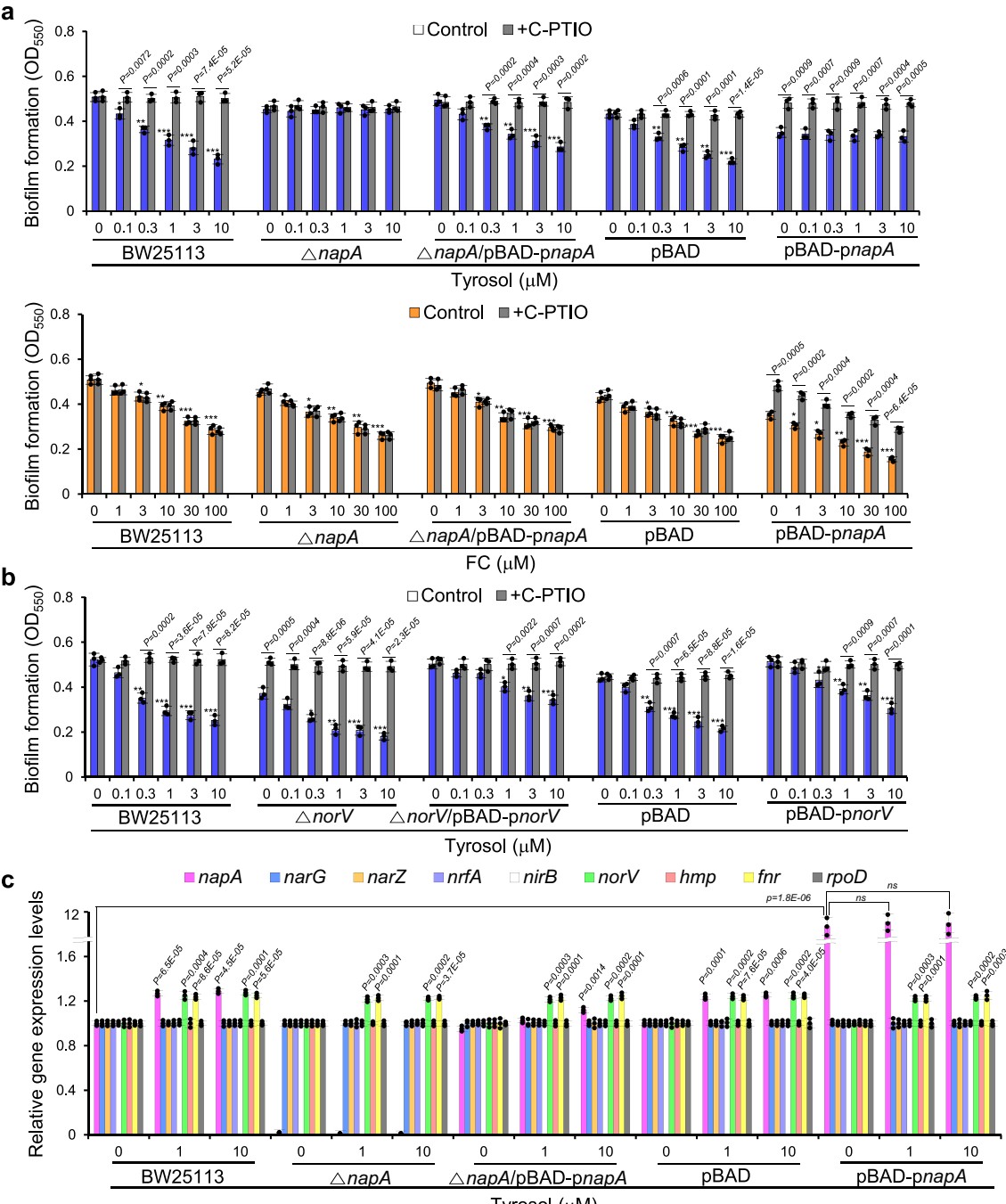

**Fig. 2 | Tyrosol-induced biofilm formation requires the periplasmic nitrate reductase (NAR)-encoding gene *napA*. a** Biofilm formation in *E. coli* BW25113, the Δ*napA* strain, the Δ*napA* strain complemented with *napA* containing a promoter region (Δ*napA* (*pBAD*-Δp*napA*)), *E. coli* BW25113 containing only a vector (pBAD), and a *napA*-overexpressing *E. coli* BW25113 strain (pBAD-p*napA*) cultured in the presence of different concentrations of tyrosol or furanone C-30 (FC) and in the absence or presence of the NO (nitric oxide) scavenger C-PTIO (2-(4-carboxyphenyl)−4,4,5,5-tetramethylimidazoline-1-oxyl-3-oxide). **b** Mediation of the effects of tyrosol by the NO reductase-encoding gene *norV*, assayed as in (**a**). **c** Effects of tyrosol on the expression of NAR-encoding genes and related genes in BW25113, the Δ*napA* strain, the Δ*napA* (*pBAD*-Δp*napA*) strain, the pBAD strain, and the pBAD-

p*napA* strain, as assessed by RT–qPCR. In **a** and **b**, the experiment shown is representative of three independent experiments, and data are presented as the mean ± SD of three independent biological replicates. *P*-values were determined using a two-sided Student's *t*-test. *, **, and *** indicate *P* < 0.01, *P* < 0.001, and *P* < 0.0001, respectively, compared to untreated cells. In **c**, the data shown are representative of three independent experiments, and data are presented as the mean ± SD of three independent biological replicates. *P*-values were determined using a two-sided Student's *t*-test and, unless otherwise indicated, data were compared with untreated cells. ns indicates not significant. Source data are provided as a Source Data file.

tyrosol failed to inhibit biofilm formation in only the *ybfA*-overexpressing *E. coli* strain, whereas FC inhibited biofilm formation in the *ybfA*-overexpressing *E. coli* strain regardless of the presence of arabinose (Figs. 3ai and S6c). Additionally, tyrosol could not inhibit c-di-GMP production and could not stimulate PDE activity but enhanced DGC

activity in the *ybfA*-overexpressing *E. coli* strain (Fig. 3aii–iv). These results indicate that YbfA is a target of tyrosol. To validate that tyrosol binds with YbfA, a direct binding assay was conducted using isothermal titration calorimetry (ITC). Tyrosol binds to YbfA in an exothermic reaction with a dissociation constant ($K_d$) of 4.65 ± 0.11 μM (Fig. 3b).

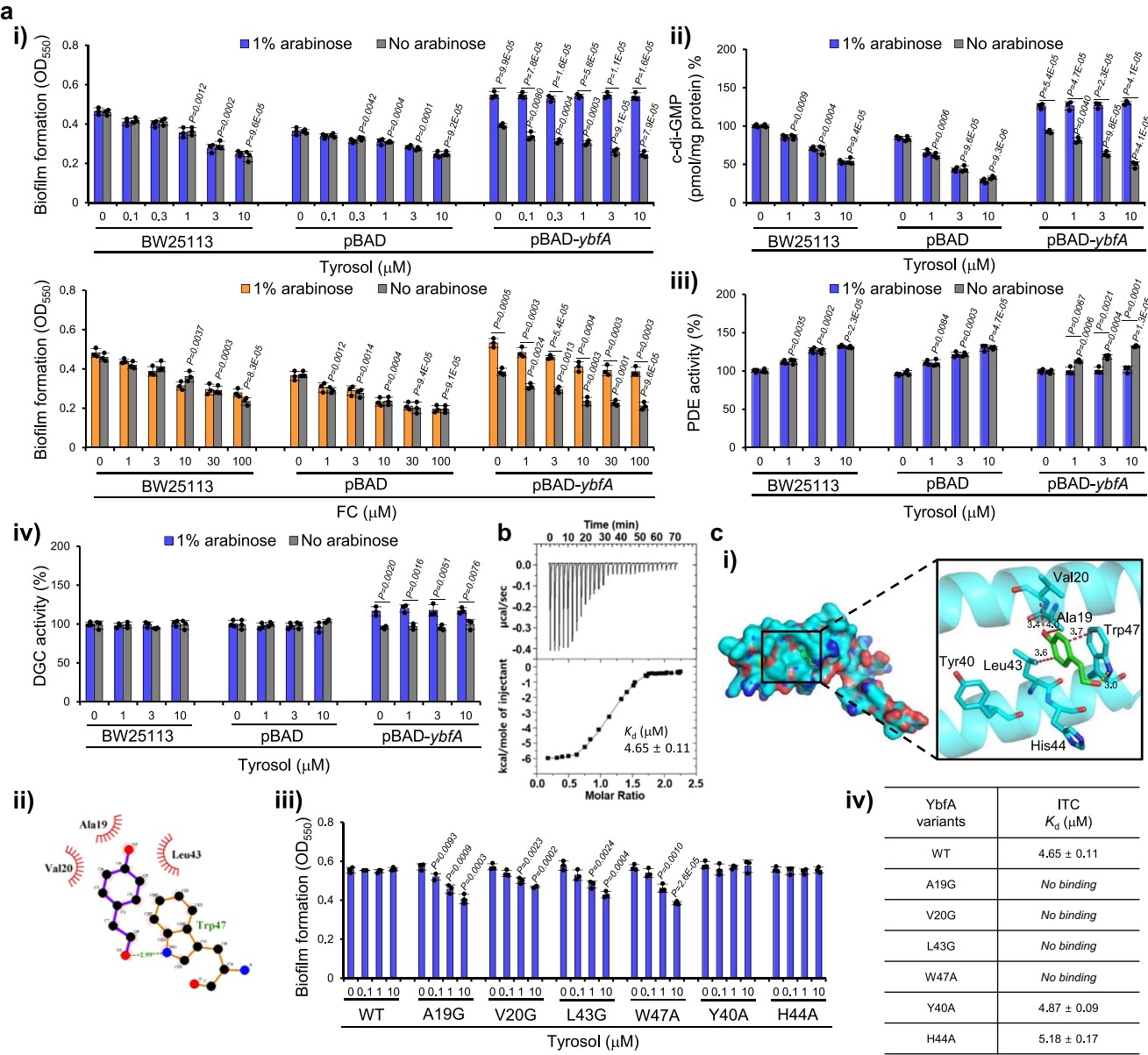

**Fig. 3 | Tyrosol inhibits biofilm formation by targeting YbfA. a** Overexpression of *ybfA* in *E. coli* reverses the biofilm formation, c-di-GMP level, and PDE activity phenotypes induced by tyrosol. Biofilm formation **i**, intracellular c-di-GMP levels **ii**, and PDE **iii** and DGC **iv** activities in *E. coli* BW25113, *E. coli* BW25113 containing only a vector (pBAD), and a *ybfA*-overexpressing *E. coli* BW25113 strain (pBAD-*ybfA*) cultured in the presence of tyrosol or furanone C-30 (FC) and in the presence or absence of arabinose. **b** Tyrosol directly binds to YbfA. Molecular interactions of YbfA and tyrosol were measured using isothermal titration calorimetry (ITC), with tyrosol being titrated into YbfA. The image shown is representative of three independent experiments with similar results. **c** Tyrosol interacts with Ala19, Val20, Leu43 and Trp47 residues of YbfA. **i** Molecular docking of tyrosol into YbfA. Tyrosol is shown in green. N and O atoms are colored blue and red, respectively. C atoms in YbfA residues and tyrosol are colored cyan and green, respectively. The hydrogen bond and representative hydrophobic interactions are depicted as

yellow and red dotted lines, respectively. The lengths of these interactions are specified in angstroms (Å). **ii** LigPlot⁺ analysis of tyrosol bound to YbfA. Hydrophobic interactions and hydrogen bond were shown as curved red and green dotted lines, respectively. **iii** In vivo binding assays of YbfA variants to tyrosol. Biofilm formation in *E. coli* BW25113 strains overexpressing YbfA variants was tested compared with the *E. coli* BW25113 strain overexpressing YbfA wild-type (WT) in the presence of tyrosol and arabinose for 18 h. **iv** The binding affinities of YbfA variants to tyrosol. The dissociation constants of YbfA variants were derived from ITC binding experiments with tyrosol (Fig. S7c). Data are presented as the mean ± SD of three independent experiments. In **a** and **ciii**, the experiment shown is representative of three independent experiments, and data are presented as the mean ± SD of three independent biological replicates. *P*-values were determined using a two-sided Student's *t*-test and, unless otherwise indicated, data were compared with untreated cells. Source data are provided as a Source Data file.

Furthermore, to gain insight into the molecular mechanism of tyrosol's binding to YbfA, key amino acid residues in YbfA interacting with tyrosol were identified using a combination of protein structure prediction, molecular docking, and site-directed mutagenesis. The 3D structure of the YbfA protein was modeled with high confidence using ColabFold, an AlphaFold2-powered and easy-to-use tool[48] (Fig. S7a). Molecular docking analysis of the predicted protein structure with the ligand (tyrosol), conducted using

AutoDock Vina[49], revealed the binding of tyrosol within a single binding pocket (Fig. 3ci). A potential hydrogen bond interaction between a hydroxyl group of tyrosol and the residue Trp47 was observed. Moreover, tyrosol was predicted to engage in hydrophobic interactions with the residues Ala19, Val20, and Leu43 (Figs. 3ci and cii). Subsequently, to determine which amino acid residues interact with tyrosol, site-directed mutants of all six amino acid residues in the binding pocket (A19G, V20G, L43G, W47A,

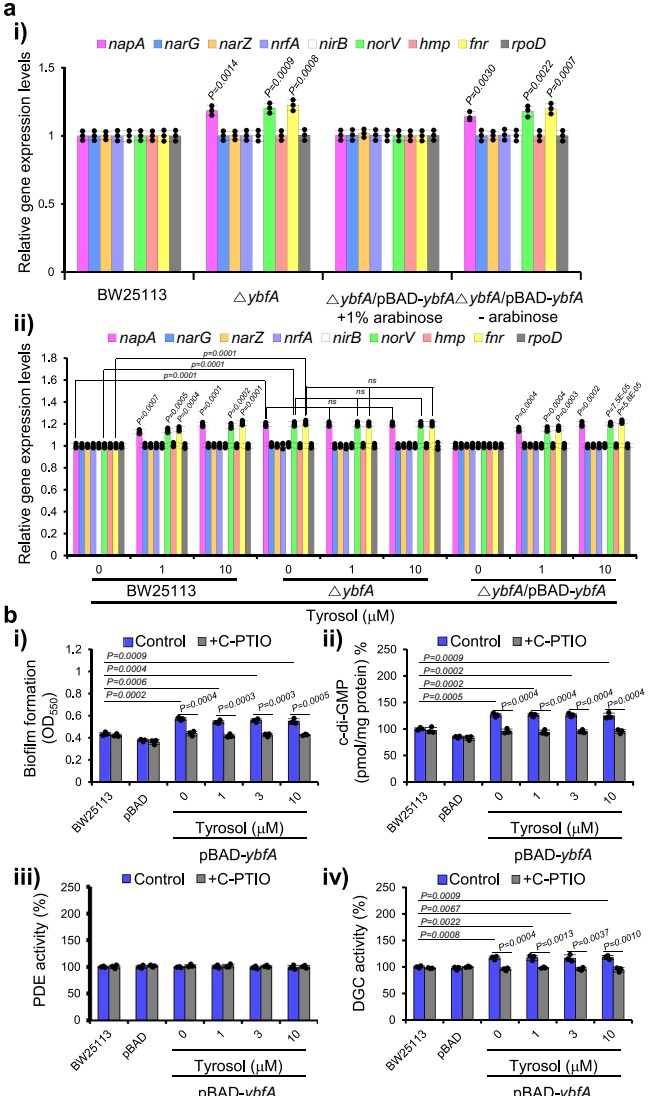

**Fig. 4 | YbfA partially suppresses transcription of nitrate reductase NapA that causes controlled NO production to activate DGC activity for production of c-di-GMP levels and subsequent stimulation of biofilm formation. a** Expression of the nitrate reductase (NAR)-encoding gene *napA* and *fnr* is elevated in the *E. coli ybfA* mutant and in the presence of tyrosol as well as suppressed by complementation with *ybfA*. **i** Expression of the NAR-encoding genes and related genes in wild-type *E. coli* BW25113, the Δ*ybfA* strain, and the Δ*ybfA* strain complemented with *ybfA* (Δ*ybfA* (*pBAD-ybf*)), as assessed by RT–qPCR. **ii** Effects of tyrosol on the expression of NAR-encoding genes and related genes in BW25113, the Δ*ybfA* strain, and the Δ*ybfA* (*pBAD-ybf*) strain, as assessed by RT–qPCR. **b** Overexpression of *ybfA* in *E. coli* induces biofilm formation, increases c-di-GMP levels, and stimulates DGC activity, and these effects are blocked by the NO (nitric oxide) scavenger C-PTIO (2-(4-carboxyphenyl)−4,4,5,5-tetramethylimidazoline-1-oxyl-3-oxide). Biofilm formation **i**, c-di-GMP levels **ii**, and PDE **iii** and DGC **iv** activities in *E. coli* BW25113, *E. coli* BW25113 containing only a vector (pBAD), and the *ybfA*-overexpressing *E. coli* BW25113 strain (pBAD-*ybfA*) cultured with tyrosol and 1% arabinose in the absence or presence of C-PTIO. In **ai**, the experiment shown is representative of three independent experiments, and data are presented as the mean ± SD of three independent biological replicates. *P*-values were determined using a two-sided Student's *t*-test and data were compared with wild-type *E. coli* BW25113. In **aii**, the experiment shown is representative of three independent experiments, and data are presented as the mean ± SD of three independent biological replicates. *P*-values were determined using a two-sided Student's *t*-test and, unless otherwise indicated, data were compared with untreated cells. ns indicates not significant. In **b**, the experiment shown is representative of three independent experiments, and data are presented as the mean ± SD of three independent biological replicates. *P*-values were determined using a two-sided Student's *t*-test. Source data are provided as a Source Data file.

being inhibited by FC, as expected (Fig. S7d). Overall, these results indicate that tyrosol interacts with the residues Ala19, Val20, Leu43, and Trp 47 in YbfA as depicted in Fig. 3ci.

These results demonstrate that tyrosol inhibits biofilm formation via NO production, increases in PDE activity and subsequent decreases in c-di-GMP levels, by targeting the stress-related protein YbfA in *E. coli*.

## YbfA partially represses NapA transcription, leading to NO production that activate DGCs for c-di-GMP synthesis

Next, after identifying YbfA as a target by which tyrosol induces the production of NO, we next aimed to clarify how YbfA affects NO production by analyzing the mRNA levels of all the genes encoding components of the denitrification pathway in *E. coli* Δ*ybfA*, the Δ*ybfA* mutant complemented with *ybfA*, and a *ybfA*-overexpressing strain by RT–qPCR. The mRNA levels of *napA*, *norV* and *fnr* were increased in the *ybfA* mutant (Fig. 4ai), which is consistent with the results in tyrosol-treated cells and thus corroborates that YbfA is a target of tyrosol. These results suggest that YbfA suppresses expression of *napA* which is required for tyrosol antibiofilm activity. This hypothesis was confirmed using the *ybfA*-complemented strain under the arabinose-inducible promoter. In the presence of arabinose, *ybfA* complementation suppressed the increase in the mRNA levels of *napA*, *norV* and *fnr* in the *ybfA* mutant (Fig. 4ai). This finding demonstrates that the poorly characterized protein YbfA suppresses *napA* transcription.

Next, to determine whether tyrosol increases *napA* transcription by inhibiting the expression of YbfA, we analyzed the mRNA levels of all the genes encoding components of the NO production pathway in *E. coli* Δ*ybfA*, the *ybfA* mutant complemented with *ybfA*, and a *ybfA*-overexpressing strain in the presence of tyrosol. The increase in the mRNA levels of *napA* in the *ybfA* mutant compared to those in wild-type *E. coli* was not affected by tyrosol treatment, as expected, but when the *ybfA* mutant was complemented with *ybfA*, tyrosol again increased the mRNA levels of *napA*, which were suppressed in the *ybfA* mutant (Fig. 4aii). Overall, these results demonstrate that YbfA suppresses *napA* transcription, and thus, tyrosol stimulates *napA* expression by blocking the ability of YbfA to repress *napA* transcription.

Y40A, and H44A) were individually created. Initially, the in vivo binding activities of the resulting YbfA variants to tyrosol were assessed by examining the effects of tyrosol, or FC as positive control, on biofilm formation in strains overexpressing YbfA variants. Mutations of Ala19, Val20, Leu43, and Trp47 to glycine (except Trp47 to alanine) resulted in the failure to abolish tyrosol's antibiofilm activity in strains overexpressing their respective *ybfA* variants (Fig. 3ciii). In contrast, mutations of His44 and Tyr40 to alanine led to a loss of tyrosol's antibiofilm activity, similar to the YbfA wild-type (WT), while FC still exhibited antibiofilm activity (Figs. 3ciii and S7b). This suggest no in vivo binding of tyrosol to the A19G, V20G, L43G and W47A variants, while binding to Y40A and H44A occurred similarly to WT. Next, the binding affinities of the YbfA variants to tyrosol were tested using ITC. Consistent with the in vivo binding assay results, tyrosol did not bind to the A19G, V20G, L43G, and W47A variants (Figs. 3civ and S7c). In contrast, it is bound to the Y40A and H44A variants, with $K_d$ values similar to that of WT. However, the likelihood that the site-directed mutations significantly alter the 3D structure of YbfA, thereby abolishing tyrosol's binding affinity to YbfA variants, can be dismissed. This is supported by the fact that these variants retained the biofilm-stimulating activity characteristic of the YbfA WT in the complementation experiments (Fig. S7d). Additionally, in line with the absence of tyrosol binding to the A19G, V20G, L43G, and W47A variants, their biofilm formation activity was not inhibited by tyrosol, while still

The YbfA-mediated repression of *napA* transcription and *ybfA* overexpression-induced enhancement of c-di-GMP production and biofilm formation compared to those of wild-type *E. coli* (pBAD) (Fig. 3ai and aii) suggested that YbfA might partially suppress *napA* transcription and that the resulting low levels of NO promote c-di-GMP production and biofilm formation. Thus, to test whether NO produced by the partially suppressed NapA induces biofilm formation, the effects of C-PTIO on c-di-GMP production and biofilm formation in the *ybfA* overexpression strain were tested. Indeed, c-di-GMP production and biofilm formation, which were enhanced by *ybfA* overexpression, returned to normal upon C-PTIO treatment (Fig. 4bi and bii). These results indicate that partial suppression of NapA activity by YbfA leads to endogenous NO production and subsequent c-di-GMP production and biofilm formation. Additionally, *ybfA* overexpression led to an increase in DGC activity but not PDE activity, and this increase was abrogated by C-PTIO treatment (Fig. 4biii and biv). These results indicate that partial suppression of the expression of the NAR NapA by YbfA decreases NO production to activate DGC activity and in turn elevate c-di-GMP levels to promote biofilm formation.

## YbfA regulates NO production and subsequent biofilm formation via the transcriptional regulator YfeR

YbfA, a predicted inner membrane protein, has been reported to respond to environmental stressors, including antimicrobial substances and radiation, but has not been well characterized[50,51]. To understand the downstream mechanism by which YbfA regulates *E. coli* biofilm formation, potential downstream targets of YbfA were investigated using differential RNA-seq and mutant analysis (Fig. 5a). We identified genes whose expression was up- or downregulated in the *ybfA* mutant compared to the wild-type strain and the *nirC* and *wzb* mutants by differential RNA-seq analysis. Because the *wzb* and *nirC* mutants exhibit phenotypes similar to the *ybfA* mutant, including reduced biofilm formation, lower c-di-GMP levels, and higher PDE activity (which returned to wild-type levels following C-PTIO treatment), as well as unchanged DGC activity compared to those of the wild type, with the exception for *nir* expression[10], these mutants were employed as controls to reduce any background effects. The expression of 91 and 122 genes was up- and downregulated by 2-fold or more, respectively, in the *ybfA* mutant compared to wild-type *E. coli* (Supplementary Data 1). Among these genes, the transcription levels of only five genes (*yfeR*, *nrdR*, *ybcM*, *alsK*, and *phoR*) with regulatory functions did almost unchanged in both the *nirC* and *wzb* mutants compared to the wild type (Supplementary Data 1), so these five genes were selected as potential targets of YbfA.

Among the five candidate targets, the expression of *yfeR*, *nrdR*, and *ybcM* was downregulated, whereas that of the other two genes, *alsK* and *phoR*, was upregulated in the *ybfA* mutant. To identify a downstream target of YbfA, mutant or overexpression strains of the five genes were assayed for biofilm formation, c-di-GMP levels, PDE and DGC activity, and transcription of genes in the denitrification pathway. Because *alsk* and *nrdR* mutants were not available in the Keio mutant library, *alsk* and *nrdR* overexpression strains were constructed. As a result, among the three mutant strains, only the *yfeR* mutant showed the same phenotype as the *ybfA* mutant, exhibiting less biofilm formation, lower c-di-GMP levels, higher PDE activity, and increased transcription of *napA*, *norV* and *fnr* compared to those of the wild type, and these differences were reversed by C-PTIO treatment (Fig. S8a). Additionally, *alsk* or *nrdR* overexpression did not affect the antibiofilm activity of tyrosol, as the tyrosol-induced reduction in biofilm formation was reversed by C-PTIO (Fig. S8b). These results suggest that YfeR, a LysR-type transcriptional regulator (LTTR), could be the target of YbfA.

This hypothesis was confirmed using the *yfeR* mutant complemented with *yfeR* and a *yfeR*-overexpressing strain. As for *napA*, because *yfeR* transcription was affected in the *ybfA* mutant, a *yfeR*

plasmid containing a -150 bp upstream region as a promoter region (p*yfeR*) was used for complementation and overexpression tests. Tyrosol indeed failed to inhibit biofilm formation of the *yfeR* mutant but inhibited that of the mutant complemented with *yfeR* containing the promoter region (Δ*yfeR*/pBAD-p*yfeR*), and this inhibitory effect was also reversed by C-PTIO treatment (Fig. 5bi). Additionally, p*yfeR* or *yfeR* overexpression (pBAD-p*yfeR* or pBAD-*yfeR*) led to loss of tyrosol antibiofilm activity, similar to the results for strains with altered *napA* expression (Figs. 5bi, S9a and b). The control FC inhibited the biofilm formation of the *yfeR* mutant, the complemented *yfeR* mutant, and the p*yfeR*-overexpressing strain, and this effect was not reversed by C-PTIO (Fig. 5bi). Additionally, the effects of tyrosol on c-di-GMP levels and PDE activity in the *yfeR* mutant, the p*yfeR* complemented mutant, and the p*yfeR*-overexpressing strain were similar to its effect on biofilm formation in these strains (Fig. S10). Furthermore, p*yfeR* or *yfeR* overexpression enhanced biofilm formation (Figs. 5bi, S10a and b) and increased c-di-GMP levels (Fig. S10a) and DGC activity (Fig. S10c), and these effects were abrogated by C-PTIO, similar to the results with *ybfA* overexpression. These data indicate that YbfA stimulates biofilm formation via YfeR.

Consistent with the results of the biofilm formation assay, the increased mRNA levels of *napA*, *norV*, and *fnr* in the *yfeR* mutant were not affected by tyrosol treatment, as shown by RT–qPCR experiments; however, these mRNA levels returned to wild-type levels upon the complementation of p*yfeR* into the *yfeR* mutant and were then increased by tyrosol treatment (Fig. 5bii). In the p*yfeR* overexpression strain, the increase in *napA*, *norV* and *fnr* transcription induced by tyrosol was abrogated, similar to loss of tyrosol antibiofilm activity as expected. Also as expected, the complementation of *yfeR* with a construct without the promoter region (Δ*yfeR*/pBAD-*yfeR*) did not restore the antibiofilm activity of tyrosol or increase *napA*, *norV*, or *fnr* mRNA levels (Fig. S9b, c), as observed for *napA*, corroborating the finding that YbfA, a target of tyrosol, affects *yfeR* transcription. Overall, these results indicate that YbfA targets the downstream transcriptional regulator YfeR to suppress *napA* transcription.

Because *yfeR* transcription was decreased in the *ybfA* mutant (Supplementary Data 1), we hypothesized that YbfA could positively regulate *yfeR* transcription. To test this hypothesis, the mRNA levels of *yfeR* were measured in the *ybfA* mutant, the *ybfA* mutant complemented with *ybfA*, and a *ybfA*-overexpressing strain by RT–qPCR. As expected, the transcription of *yfeR* was decreased in the *ybfA* mutant but restored to normal upon complementation of *ybfA* in the presence, but not the absence, of arabinose (Fig. 6a). Furthermore, overexpression of *ybfA* led to increases in *yfeR* mRNA levels in the presence of arabinose. To verify the binding of YbfA to the *yfeR* promoter, an electrophoretic mobility shift assay (EMSA) was performed. YbfA was observed to bind to the *yfeR* promoter DNA (P*yfeR*) (Fig. 6b). Notably, the amount bound increased with higher concentrations of YbfA (1, 3, and 10 nM), but decreased in the presence of increasing concentrations of unlabeled P*yfe*R (50, 100, and 200-fold). In contrast, the *fnr* promoter DNA (P*fnr*), used as a negative control, did not bind to YbfA, confirming the specificity of YbfA's binding to the *yfeR* promoter.

These data indicate that YbfA activates transcription of the transcriptional regulator YfeR for partial suppression of *napA*, which then regulates NO production and subsequent biofilm formation.

## Negative regulation of FNR by YfeR contributes to the repression of napA transcription

FNR has been reported to play a role as a transcriptional activator or repressor in a variety of pathways[52,53]. In Fig. 5bii, it is suggested that YfeR, which is upregulated by YbfA, represses the transcription of *fnr* and *napA*. Considering it has been reported that FNR induces *nir* or *nap* transcription in the denitrification pathway of *E. coli*[54,55], this result suggests that YfeR can repress *napA* transcription via FNR.

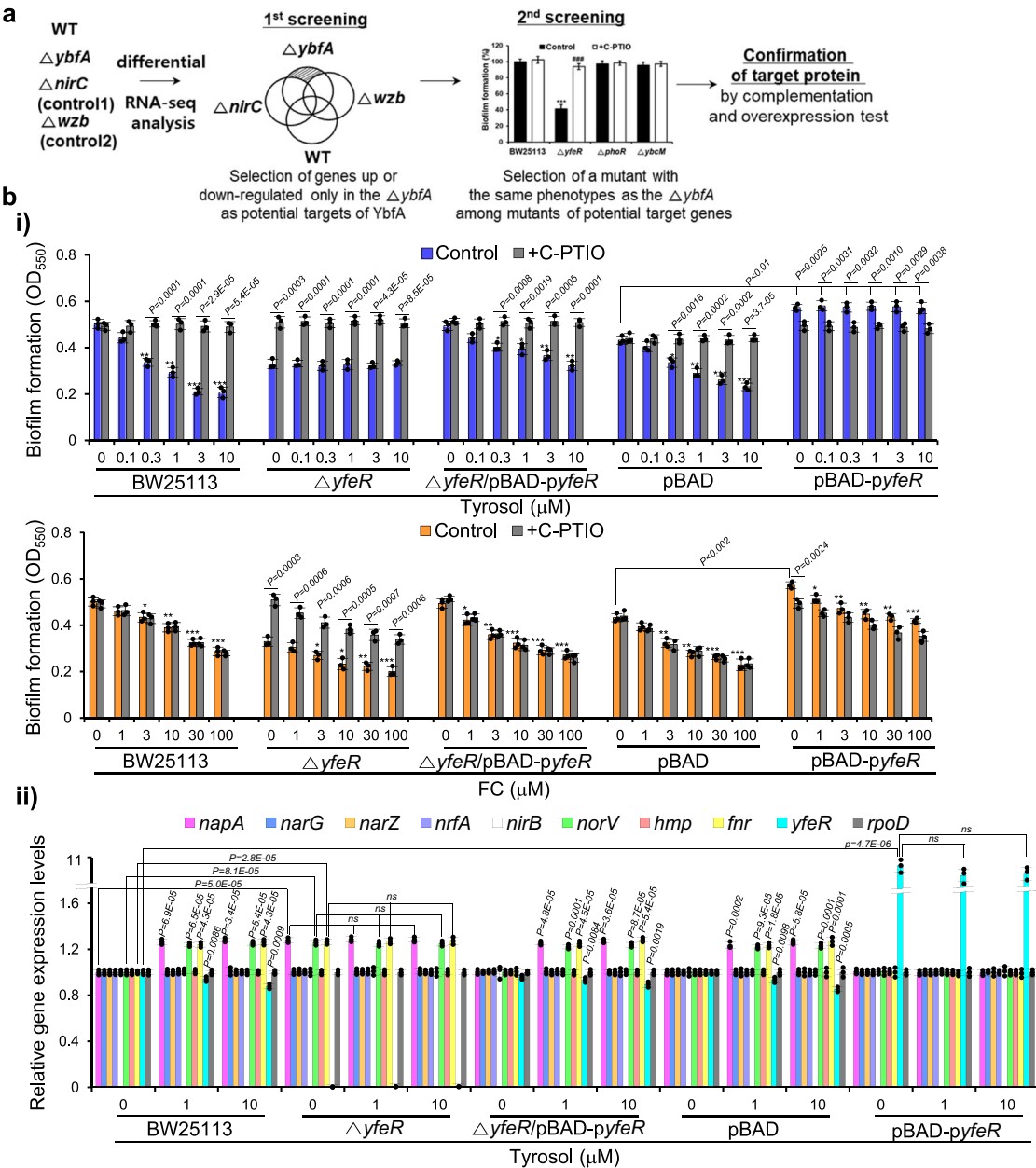

**Fig. 5 | YbfA regulates NO production and subsequent biofilm formation via the transcriptional regulator YfeR. a** Schematic strategy for identifying a downstream target of YbfA. **b** Complementation and overexpression of *yfeR* containing a promoter region reverses the biofilm formation and *napA* transcription phenotypes in a *yfeR* mutant and wild-type *E. coli*, respectively, induced by tyrosol. **i)** Biofilm formation in *E. coli* BW25113, the Δ*yfeR* strain, the Δ*yfeR* strain complemented with *yfeR* containing a promoter region (Δ*yfeR* (*pBAD*-Δp*yfeR*)), *E. coli* BW25113 containing only a vector (pBAD), and a *yfeR*-overexpressing *E. coli* BW25113 strain (pBAD-p*yfeR*) cultured in the presence of different concentrations of tyrosol and in the absence or presence of the NO (nitric oxide) scavenger C-PTIO

(2-(4-carboxyphenyl)-4,4,5,5-tetramethylimidazoline-1-oxyl-3-oxide). **ii** Effects of tyrosol on the expression of NAR-encoding genes and related genes in BW25113, the Δ*yfeR* strain, the Δ*yfeR* (*pBAD*-Δp*yfeR*) strain, the pBAD strain, and the pBAD-p*yfeR*, strain, as assessed by RT–qPCR. In **bi** and **ii**, the experiment shown is representative of three independent experiments, and data are presented as the mean ± SD of three independent biological replicates. *P*-values were determined using a two-sided Student's *t*-test and, unless otherwise indicated, data were compared with untreated cells. *, **, and *** indicate *P* < 0.01, *P* < 0.001, and *P* < 0.0001, respectively, compared to untreated cells. ns indicates not significant. Source data are provided as a Source Data file.

To verify the transcriptional regulation of FNR by YfeR, we investigated YfeR's binding to the *fnr* promoter using EMSA experiments. These experiments showed that YfeR bound to P*fnr*. The binding intensity increased with rising concentrations of YfeR at 1, 3, and 10 nM, but decreased with the addition of increasing concentrations of unlabeled P*fnr* at 50, 100, and 200-fold dilutions (Fig. 7a). As a negative control, P*yfe*R did not bind to YfeR as expected. This result confirms the specific binding of YfeR to the *fnr* promoter.

Next, to demonstrate that FNR represses *napA* transcription, the expression of individual genes in the denitrification pathway was investigated in the *fnr* mutant, the *fnr* mutant complemented with *fnr* containing a -150 bp upstream region (p*fnr*), and an *fnr*-over-expressing strain by RT–qPCR (Fig. 7bi). The mRNA levels of *napA* and *norV* were increased in the *fnr* mutant but restored to the normal levels in the wild-type strain upon complementation of the *fnr* mutant with p*fnr*. The ability of tyrosol treatment to increase *napA* and *norV* transcription was repressed by p*fnr* (Fig. 7bi)

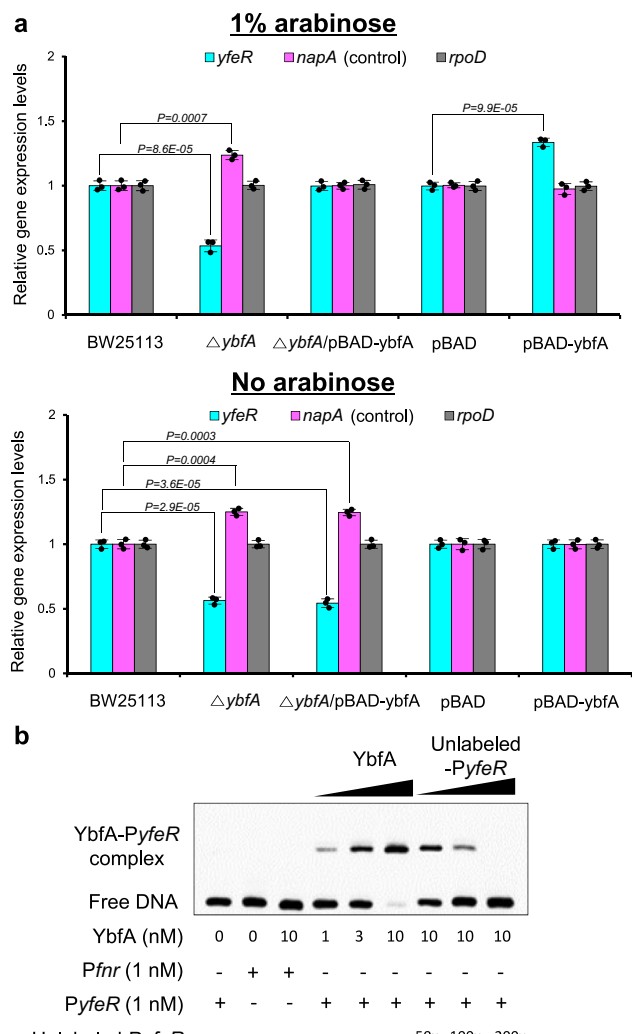

**Fig. 6 | YbfA positively regulates *yfeR* transcription in *E. coli* biofilms.**
**a** Expression of *yfeR* in biofilms of *E. coli* BW25113, the Δ*ybfA* strain, the Δ*ybfA* strain complemented with *ybfA* (Δ*ybfA* (pBAD-*ybfA*)), *E. coli* BW25113 containing only a vector (pBAD), and the *ybfA*-overexpressing *E. coli* BW25113 strain (pBAD-*ybfA*) cultured in the absence or presence of arabinose, as assessed by RT–qPCR. *napA* expression was used as a positive control. The experiment shown is representative of three independent experiments, and data are presented as the mean ± SD of three independent biological replicates. *P*-values were determined using a two-sided Student's *t*-test. **b** EMSA analysis demonstrates YbfA binding to the *yfeR* promoter. The biotin-labeled 150-bp *yfeR* promoter DNA (P*yfeR*) was incubated with increasing amounts of YbfA protein. The biotin-labeled 150-bp *fnr* promoter DNA (P*fnr*) was used as a negative control. Binding specificity was confirmed by competitive assays with increasing excesses of unlabeled P*yfeR*. The image shown is representative of three independent experiments with similar results. Source data are provided as a Source Data file.

overexpression, exactly as observed for YfeR in Figs. 5bii, which indicates suppression of *napA* transcription by FNR. The increased mRNA levels of *napA* and *norV* in the *fnr* mutant were not affected by tyrosol, but they returned to normal upon p*fnr* complementation and were further elevated by tyrosol treatment. p*fnr* overexpression abrogated the ability of tyrosol to enhance *napA* and *norV* transcription. This result indicates that tyrosol enhances *napA* transcription via FNR. Additionally, *yfeR* transcription levels, which were decreased in the *ybfA* mutant, were not affected in the *fnr* mutant, and p*fnr* overexpression did not abrogate the ability of tyrosol to reduce *yfeR* transcription (Fig. 7bi), corroborating our finding that FNR is a downstream target of YfeR. Overall, these results indicate

that FNR suppresses *napA* transcription, and tyrosol enhances *napA* transcription via FNR.

These above results suggested reduced biofilm formation in the *fnr* mutant which could not be affected by tyrosol. Indeed, the *fnr* mutant showed less biofilm formation than the wild-type strain, which was not affected by tyrosol treatment but recovered to normal levels upon C-PTIO treatment or complementation of the *fnr* mutant with p*fnr*. This restoration of biofilm formation was inhibited by tyrosol treatment but again reversed with C-PTIO (Fig. 7bii). FC, used as a control, inhibited the biofilm formation of the three mutants.

Overall, these results demonstrate that FNR, repressed by YfeR, partially suppresses *napA* transcription, resulting in low levels of NO production, enhanced DGC activity, increased c-di-GMP levels and subsequent stimulated biofilm formation. Tyrosol inhibits biofilm formation via this mechanism by targeting YbfA.

## YbfA-mediated biofilm formation is activated during the maturation-I stage of biofilm development

The anaerobic transcriptional regulator FNR becomes active state in the absence of oxygen[53]. Considering that oxygen is limited within the interior of mature biofilms[28,31], it was suggested that YbfA-mediation biofilm formation via FNR might occur during the mature stage of biofilm formation. To test this hypothesis, we assessed the biofilm formation of the *fnr* and *ybfA* mutants in comparison to the wild type over the course of biofilm development to determine the functional timing of FNR and YbfA. The *nirC* mutant, which has a biofilm-inhibition phenotype similar to that of the *ybfA* mutant but with unchanged *fnr* expression[10], was used as a negative control. Indeed, the *fnr* and *ybfA* mutants exhibited similar biofilm formation up to 6 h into biofilm development compared to the wild type, but displayed reduced biofilm formation after 9 h of biofilm development (Fig. 8a). In contrast, the *nirC* mutant, as a control, exhibited reduced biofilm formation after 1 h of biofilm development. This result suggests that FNR and YbfA operate during the late stages of biofilm formation. This notion was further confirmed by assessing the mRNA levels of *napA* in the *fnr* and *ybfA* mutants over the course of biofilm development to elucidate the timing of *napA* repression by FNR and YbfA. Flagella synthesis-associated genes (*fliA*, *flgC*, and *flgN*) and anaerobic metabolism-associated genes (*hycF*, *hycI*, and *hyaA*) were used as markers for the early stages (attachment) and late stages (maturation) of biofilm formation, respectively[56]. In our *E. coli* biofilm system, the expression of *fliA*, *flgC*, and *flgN* was substantial at planktonic cells, peaked at 1 h-old biofilms, and began to decline at 3 h-old biofilms (Fig. 8bi). In contrast, the expression of *hycF*, *hycI*, and *hyaA* was minimal at both the planktonic cells and 1 h-old biofilms, substantial at 6 h-old biofilms, and reaching its peak at 12 h-old biofilms (Fig. 8bi). This result suggested that biofilms aged 6 h and 12 h are at the maturation-I and −II stages, respectively. Consistent with the expression profiles of *hycF*, *hycI*, and *hyaA*, *napA* expression in the *fnr* and *ybfA* mutants was minimal at 1 h-old biofilms and peaked at 12 h-old biofilms (Fig. 8bii). In contrast, *nirB* transcription in the *nirC* mutant, as a control, was highest at the planktonic cells. Importantly, the expression of *hycF*, *hycI*, and *hyaA* was decreased in the *fnr* and *ybfA* mutants (Fig. 8bi), consistent with the decrease in their biofilm formation, whereas was not changed in the *nirC* mutant as a control. This result supports that YbfA and FNR mediate biofilm formation under anaerobic conditions.

Overall, these results indicate that YbfA and FNR function during the maturation-I stage of biofilm formation in which oxygen is limited inside biofilms.

## Blocking YbfA increases susceptibility of biofilm cells to antibiotics

Mature biofilms are reportedly resistant to antimicrobials due to oxygen limitation within their interior[28,31]. Consistent with the findings that tyrosol inhibits the maturation-I stage of biofilm development

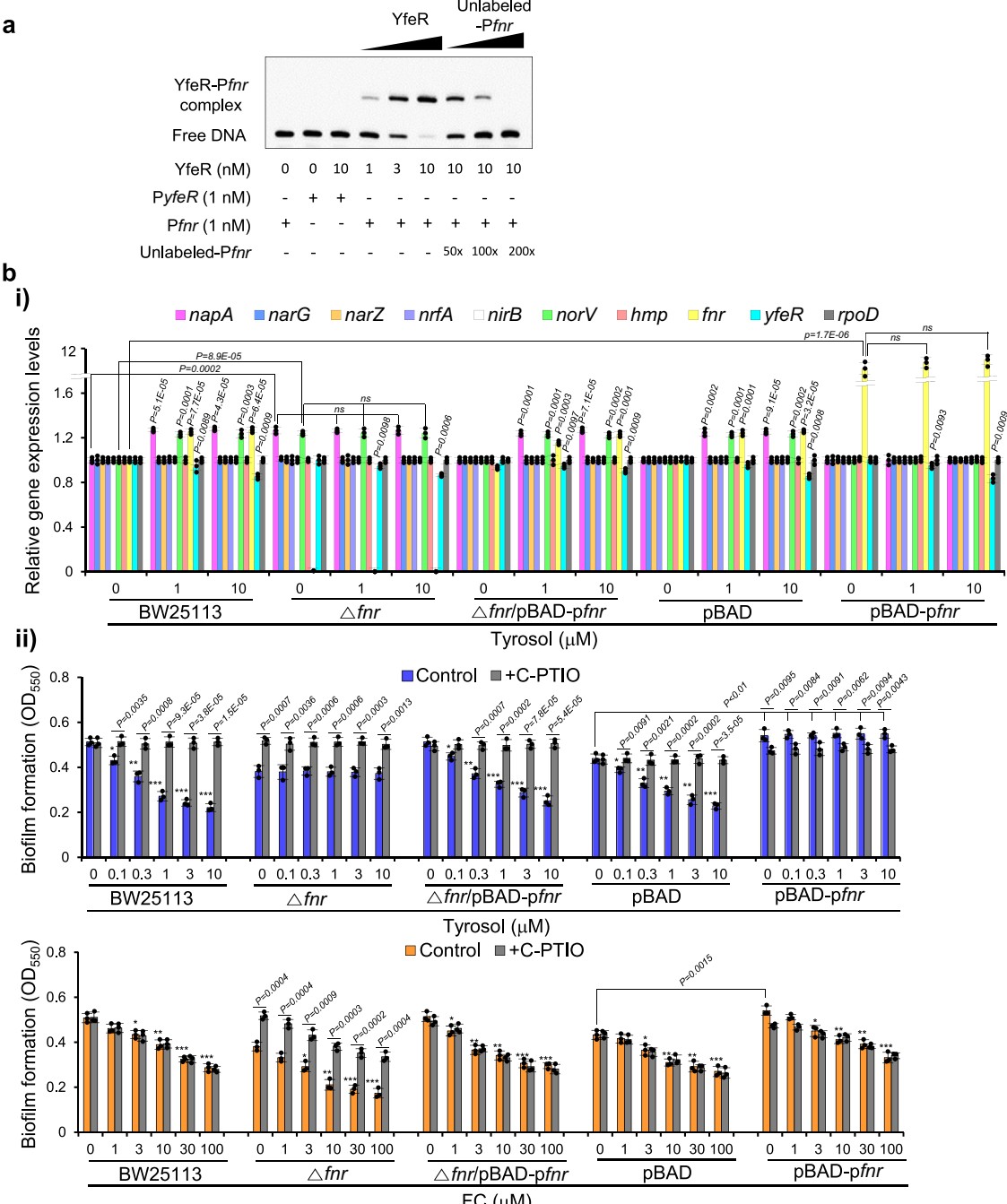

**Fig. 7 | Negative regulation of FNR by YfeR contributes to the repression of *napA* transcription. a** EMSA analysis indicates YfeR binding to the *fnr* promoter. The biotin-labeled P*fnr* DNA was incubated with increasing amounts of YfeR protein. The biotin-labeled P*yfeR* DNA served as a negative control. Competitive assays were performed with increasing excesses of unlabeled P*fnr*. The image shown is representative of three independent experiments with similar results.
**b** Complementation and overexpression of *fnr* containing a promoter region reverses the biofilm formation and *napA* transcription phenotypes in an *fnr* mutant and wild-type *E. coli*, respectively, induced by tyrosol. **i** Effects of tyrosol on the expression of NAR-encoding genes and related genes in biofilms of BW25113, the Δ*fnr* strain, the Δ*fnr* (*pBAD*-p*fnr*) strain, the pBAD strain, and the pBAD-p*fnr* strain, as assessed by RT–qPCR. **ii** Biofilm formation in *E. coli* BW25113, the Δ*fnr* strain, the

Δ*fnr* strain complemented with *fnr* containing a promoter region *(Δfnr (pBAD-fnr))*, *E. coli* BW25113 containing only a vector (pBAD), and an *fnr*-overexpressing *E. coli* BW25113 strain (pBAD-p*fnr*) cultured in the presence of different concentrations of tyrosol or furanone C-30 (FC) and in the absence or presence of the NO (nitric oxide) scavenger C-PTIO (2-(4-carboxyphenyl)−4,4,5,5-tetramethylimidazoline-1-oxyl-3-oxide). In **bi** and **bii**, the experiment shown is representative of three independent experiments, and data are presented as the mean ± SD of three independent biological replicates. *P*-values were determined using a two-sided Student's *t*-test and, unless otherwise indicated, data were compared with untreated cells. ns indicates not significant. *, **, and *** indicate *P* < 0.01, *P* < 0.001, and *P* < 0.0001, respectively, compared to untreated cells. Source data are provided as a Source Data file.

through YbfA and FNR, the expression of genes associated with anaerobic metabolism was decreased following tyrosol treatment (Fig. 8c). This result suggests that biofilm cells developed in presence of tyrosol may exhibit reduced susceptibility to antibiotics. This

hypothesis was tested by investigating the effect of tyrosol on the susceptibility of biofilms cells to antibiotics. The *nirC* mutant, which did not affect the expression of genes associated with anaerobic metabolism (Fig. 8bi), along with complestatin, an inhibitor of NirC[10],

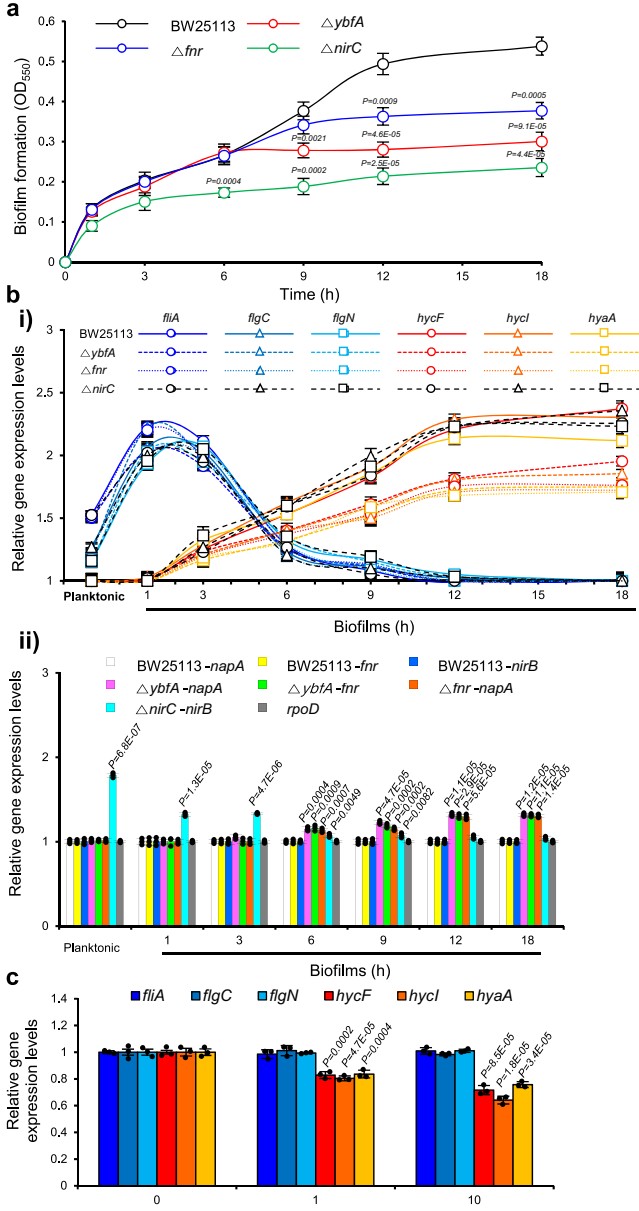

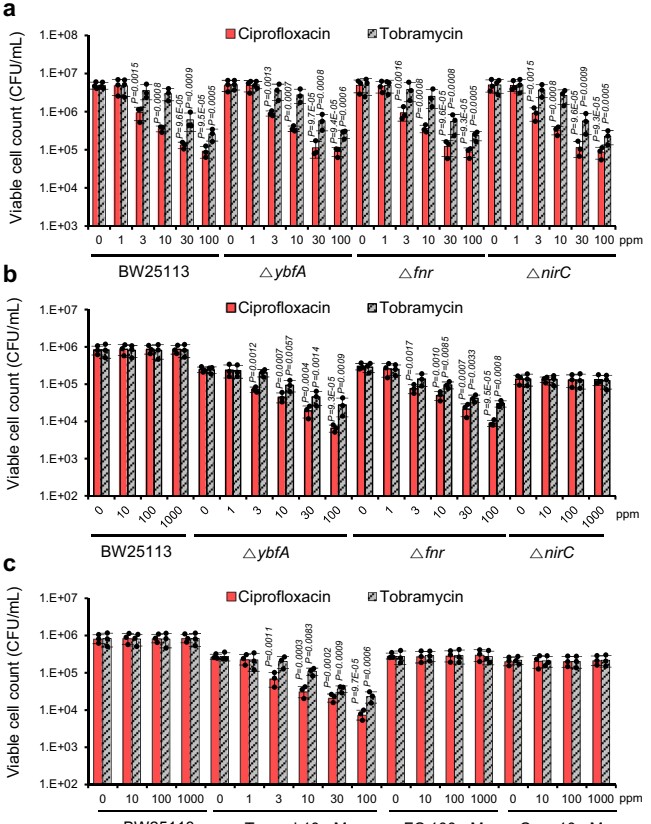

**Fig. 9 | Blocking YbfA via tyrosol treatment or through *ybfA* and *fnr* mutations increases the susceptibility of biofilm cells to antibiotics. a, b** Antibiotic susceptibility of planktonic (**a**) and biofilm (**b**) cells of wild-type *E. coli* BW25113, and of the *ybfA* and *fnr* mutants to ciprofloxacin and tobramycin. **c** Effects of tyrosol on the susceptibility of *E. coli* BW25113 biofilms to ciprofloxacin and tobramycin. Furanone C-30 (FC) and complestatin (com) were used as controls. The experiment shown is representative of three independent experiments, and data are presented as the mean ± SD of three independent biological replicates. *P*-values were determined using a two-sided Student's *t*-test and data were compared with untreated cells. Source data are provided as a Source Data file.

**Fig. 8 | Ybf-mediated biofilm formation via FNR is triggered during the maturation stage-I of biofilm formation. a** Comparison of biofilm formation by the *ybfA* and *fnr* mutants with that of the wild-type BW2511 over the course of biofilm development. The *nirC* mutant was included as a control. **b** Biofilm formation mediated by YbfA via FNR is activated during the late stage of biofilm formation. Expression of **i** flagella synthesis-related genes (*fliA*, *flaC*, and *flgN*) and anaerobic metabolism-associate genes (*hyaA*, *hycI*, and *hycF*), which are employed as marker genes for the early and late stages, respectively, of biofilm formation[56], and **ii** NO-synthesis genes in *E. coli* BW25113 and the *ybfA* and *fnr* mutants were assessed over the course of biofilm development using RT–qPCR. Cells were obtained under planktonic growth conditions for 12 h and after 1 h, 3 h, 6 h, 9 h, 12 h, and 18 h of biofilm formation. **c** Effects of tyrosol on the expression of flagella synthesis-related genes and anaerobic metabolism-associate genes in *E. coli* biofilms, as assessed by RT–qPCR. In **a**, **bii**, and **c**, the experiment shown is representative of three independent experiments, and data are presented as the mean ± SD of three independent biological replicates. *P*-values from all data were determined using a two-sided Student's *t*-test and data were compared with wild-type *E. coli* BW25113 in (**a**), planktonic wild-type *E. coli* BW25113 in **bii**, and untreated cells in **c**. Source data are provided as a Source Data file.

were used as controls. First, we compared the susceptibility of planktonic and biofilm cells of wild-type *E. coli* to bactericidal agents such as ciprofloxacin and tobramycin. While the planktonic cells of the wild type were vulnerable to both ciprofloxacin and tobramycin, showing reductions of 80.8% and 87.3% at 3 and 30 μg/mL, respectively (Fig. 9a), biofilm cells showed high resistance even up to 1000 μg/mL (Fig. 9b). Surprisingly, biofilm cells developed in presence of tyrosol (10 μM) exhibited remarkable sensitivity to ciprofloxacin and tobramycin treatment, with reductions of 74.2% and 86.5% at 3 and 30 μg/mL, respectively (Fig. 9c). In contrast, biofilm cells developed in presence of FC or complestatin remained highly tolerant to the antibiotics even up to 1000 μg/mL. This hypothesis was further validated by assessing the susceptibility of the *ybfA* and *fnr* mutant biofilms to antibiotics. The biofilm cells of both the *ybfA* and *fnr* mutants showed sensitivities to the antibiotics comparable to those observed in planktonic cells (Fig. 9a, b). As a control, the biofilm cells of the *nirC* mutants were resistant to the antibiotics, similar to the wild-type, as expected. Furthermore, the ability of tyrosol to enhance the susceptibility of biofilm cells to antibiotics was demonstrated in several clinical isolates of both *E. coli* and *P. aeruginosa* (Figs. S11 and 12).

Overall, these findings strongly indicate that tyrosol inhibits biofilm formation during the early maturation stage through targeting YbfA and FNR, thereby preventing the development of anaerobic

biofilms, and consequently increasing the susceptibility of biofilm cells to antibiotics.

## Discussion

NO plays a dual role in biofilm processes—both in formation and dispersion, creating a paradox. While the dispersal mechanism linked to NO generated by NIR is relatively understood[1,9], the NO-induced biofilm formation mechanism, especially how low levels of NO is regulated for biofilm formation, remains largely unexplored. Here, we aimed to clarify the downregulation of NO production to facilitate biofilm formation by investigating inhibitors of biofilm formation that stimulate NO production. We found that tyrosol, a safe natural product[38], exerted potent antibiofilm activity by enhancing NO production (Fig. 10). This anti-biofilm activity was remarkably more effective, tenfold, than that of FC, a widely recognized QS inhibitor, in *E. coli* and *P. aeruginosa*. We also found that tyrosol produced NO via stimulation of the transcription of the periplasmic NAR NapA. This differs from the mechanism involving complestatin's NirB[10]. This discovery led us to identify the target protein of tyrosol. In fact, through analysis of an *E. coli* mutant library, we identified the poorly characterized protein YbfA as the target of tyrosol. Our findings demonstrated that YbfA stimulated biofilm formation by maintaining low levels of NO via partial suppression of *napA* transcription. Additionally, we revealed that YbfA partially suppressed *napA* transcription via the sequential functions of YfeR and interestingly the anaerobic transcriptional regulator FNR; YbfA activates yfeR *transcription*, and then YfeR suppresses *fnr* transcription, resulting in the partial suppression of *napA* transcription. Consistent with the role of the anaerobic regulator FNR, YbfA-mediated biofilm formation via FNR was demonstrated to activate during the maturation-I stage of biofilm development, where biofilms form under anaerobic conditions. Of great significant, blocking YbfA either through tyrosol treatment or *ybfA* mutation increased sensitivity of biofilm cells to antibiotics such as ciprofloxacin and tobramycin, making them nearly as susceptible as planktonic cells. This study reveals a mechanism by which YbfA mediates biofilm formation and maintains low NO production via YfeR and FNR. Importantly, tyrosol enhances biofilm cell sensitivity to antibiotics through the mechanism.

In this study, YbfA facilitates biofilm formation by triggering a controlled production of NO via the sequential regulation of YfeR and FNR. YbfA reportedly is associated with stress response[50,51], but has not been well characterized. The *ybfA* mutant of *E. coli* K12 has shown reduced sensitivity to antimicrobial peptide plantaricin BM-1[51] and increased susceptibility to X-ray and UV radiation[50], indicating its stress-related role. This study reveals the enhanced sensitivity of the *ybfA* mutant to ciprofloxacin and tobramycin via YfeR and FNR in anaerobic biofilms. This is different from the reduced sensitivity of the *ybfA* mutant of *E. coli* K12 to plantaricin, a membrane-targeting bacteriocin, mediated via the BasS/BasR two-component systems in aerobically grown planktonic culture[51]. This suggests that YbfA may affect the sensitivity of *E. coli* to antibiotics differently depending on the oxygen conditions or the type of antibiotic used. Notably, this study reveals that YbfA-mediated biofilm formation is activated under oxygen limitation conditions, which represent a form of stress. This finding underscores a potential link between YbfA-mediated biofilm formation and stress tolerance. These insights illuminate how bacteria adapt to environmental changes, particularly limited oxygen conditions. This understanding could be crucial not only for treating bacterial infections but also for broader applications in biotechnology, wastewater treatment, and other fields where biofilms play a significant role.

This study reveals that YfeR is a downstream target of YbfA. YfeR, although structurally categorized within the LTTR family, has remained relatively unexplored, except for its role in optimizing growth under low-osmolarity conditions in *S. typhimurium*[57]. LTTRs

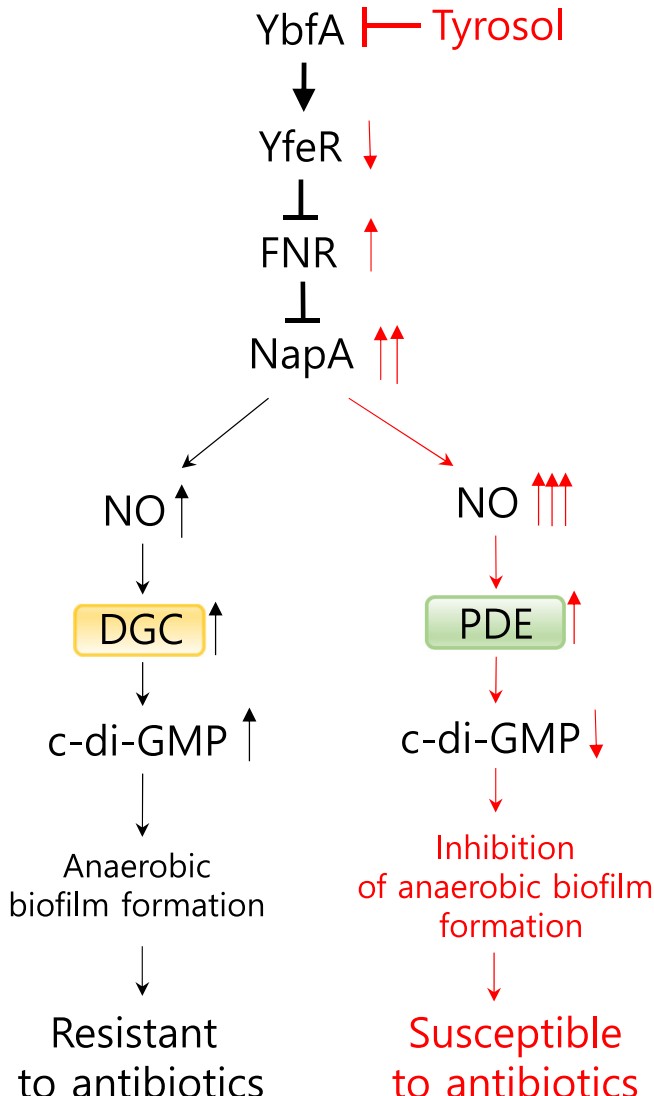

**Fig. 10 | The hypothetical protein YbfA contributes to anaerobic biofilm formation by decreasing NO production via YfeR, FNR and NapA.** During the late stages of biofilm development under oxygen limited conditions, YbfA activates *yfeR* transcription, which in turn suppresses *fnr* transcription. Low levels of NO (nitric oxide), resulting from FNR suppression of *napA* transcription, activate DGC activity and lead to elevation of c-di-GMP levels and subsequent biofilm formation. Tyrosol inhibits biofilm formation by targeting YbfA in the following sequential process: deactivation of YfeR, subsequent release from FNR suppression, release from NapA suppression, high levels of NO production, enhanced PDE activity, decreased c-di-GMP levels, and reduced anaerobic biofilm formation. This decrease in anaerobic biofilm formation renders biofilm cells susceptible to antibiotics. A flat red arrowhead indicates the target-inhibition of YbfA by tyrosol. Solid arrowhead represent positive transcriptional regulation, while flat arrowheads denote negative transcriptional regulation.

include regulatory proteins that activate or repress the expression of genes encoding proteins related to virulence, metabolism, QS and motility in response to environmental stimuli[58,59]. Some LTTRs like LeuO[60], OsaR[61], and PA2121[62] have been reported to control biofilm formation negatively in pathogens like *A. baumanii* or *P. aeruginosa*. In this study, unusually for other LTTRs, YfeR promotes biofilm formation by orchestrating controlled endogenous NO levels through FNR and NapA. Thus, this finding paves the way for further investigations into other potential members of the LTTR family in biofilm formation of different organisms.

In this study, the global transcriptional regulator FNR was found to be a downstream target of YfeR. In the denitrification pathway for planktonic growth under anaerobic conditions in *E. coli*, FNR reportedly induces the transcription of *nir*[54], *nrf*[55,63], and *nap*[55]. This study reveals the suppression of transcription of *napA*, not *nirB*, by YfeR-downregulated FNR during anaerobically biofilm formation, which is different from FNR regulation of *nap*, *nirB*, and *nrf* transcription during anaerobically planktonic growth. Thus, this study illuminates FNR's involvement in *E. coli* biofilm formation. Furthermore, while the downstream target genes of FNR are well-studied within the denitrification pathway[53,64], the mechanisms upstream of FNR remain unexplored. This study, by discovering YfeR as an upstream regulator of FNR, provides insights into how FNR activity might be modulated in other FNR-mediated processes.

This study shows that the anaerobic regulator FNR is essential for YbfA-mediated biofilm formation. FNR, featuring a Fe-S cluster in the sensory domain, becomes an active state in the absence of oxygen[53]. This implied that the mechanism driving YbfA-mediated biofilm formation might be initiated after the formation of appropriate biofilms, creating an anaerobic environment. Local depletion of oxygen within the biofilm has been evidenced using oxygen microelectrode technology, subsequently restricting protein synthesis within the interior of mature *P. aeruginosa* biofilms, causing antibiotic resistance[28]. Reportedly, oxygen limitation is responsible for a minimum of 70% of the antibiotic resistance in mature *P. aeruginosa* biofilm cells[31]. Oxygen deficiency also increases the antibiotic resistance of *E. coli*[65]. In this study, YbfA-mediated biofilm formation via FNR is triggered during the maturation-I stage of biofilm development under oxygen-limited conditions (Fig. 8). In contrast, NirC-mediated biofilm formation is initiated during the early stage under aerobic conditions (Fig. 8). These findings are consistent with the results that elevated transcription of *fnr* and *napA* in tyrosol-treated cells (Fig. 1d) were detected in biofilm cells, not planktonic cells, unlike the case of *nirB* in complestatin's NirC[10]. A decreased anaerobic metabolism in the *ybfA* mutant compared to the wild type or the *nirC* mutant (Fig. 8i) suggests a reduced anaerobic biofilm formation in the *ybfA* mutant. This could explain why inhibiting YbfA through tyrosol treatment or *ybfA* mutation makes biofilms cells dramatically susceptible to antibiotics, while blocking NirC has no such effect. While the exact mechanism by which YbfA is activated during oxygen-limited conditions remains to be elucidated, this study offers new perspectives on strategies to overcome antibiotic resistance posed by oxygen-depleted biofilm environments in treating bacterial infections.

In this study, nitrate reductase NapA is responsible for NO production in tyrosol-induced biofilm inhibition. Nitrate reductase NARs (*nap* and *nar*) catalyze the conversion of nitrate to nitrite, a substrate of NO, while nitrite reductases NIR (*nir* and *nrf*) produce NO from nitrite[18]. How does NapA contribute to NO production? In this study, all denitrification genes and related genes, including the nitrite reductase-coding genes (*nirB* and *nrfA*) were expressed in the untreated biofilm cells as a control (Fig. 1f), which is consistent with the requirement of low levels of NO for biofilm formation[10]. The expression of *napA*, along with *norV* and *fnr*, was upregulated in tyrosol treatment or in the *ybfA* mutant, compared to the untreated cells. Thus, the upregulation of *napA*, whether due to *ybfA* mutation or tyrosol treatment, is suggested to contribute to increased production of NO by enhancing the production of nitrite, a substrate of NIR.

In this study, NorV does not affect tyrosol-mediated biofilm inhibition. This result is consistent with the known NO-eliminating role of NorV, whose encoding gene is induced by NO[44,45]. The elimination of excess NO by NorV during biofilm formation is supported by findings that biofilm formation of the *norV* mutant was reduced compared to that of the wild type, and these effects were reversed by C-PTIO treatment (Fig. 2b). It was also demonstrated that the decreased biofilm formation of the *norV* mutant was restored to the levels of the wild-type strain through complementation with p*norV* (Fig. 2b). These results suggest that excess toxic NO, which inhibits biofilm formation regardless of tyrosol treatment, is eliminated by NorV. Although three proteins, including a cytochrome *c* nitrate reductase (NrfA), flavohemoglobin (Hmp), and flavorubredoxin (NorV), are known as NO-detoxifying enzymes in *E. coli*[45], this study highlights NorV as a significant contributor to NO homeostasis in this system.

Additionally, *norV* expression is regulated by FNR (Fig. 7bi). Reportedly, in response to NO, the NO-response transcriptional regulator NorR activates *norV* transcription[64], whereas FNR regulates *hmp* or *nrf* transcription in the denitrification process in *E. coli*[66]. Thus, unlike *norV* regulation by NorR in the denitrification process, *norV* expression in *E. coli* biofilm formation is regulated by FNR. Overall, these results indicate that NorV, regulated by FNR, contributes to biofilm formation by eliminating excess NO in *E. coli*. This finding sheds light on the role of NorV in maintaining proper levels of NO for biofilm formation.

This study reveals that tyrosol showed potent antibiofilm activity against *E. coli* as well as *P. aeruginosa* by targeting YbfA. Tyrosol is a safe natural product present in edible plant products such as olive oil and wine. Tyrosol, one of the polyphenols, is reported to have various beneficial effects on human health by preventing hypertension, atherosclerosis, chronic heart failure, insulin resistance, ischemia-related stress, etc .[67,68]. The inhibitory effects of tyrosol on QS, biofilm formation, and virulence factor production in *P. aeruginosa* have been reported[69,70]. Although the antibiofilm activity of tyrosol on *Streptococcus mutans* and *Candida albicans* has also been reported, its exact mechanism remains unknown[71]. Thus, the potent antibiofilm activity of tyrosol, mediated by targeting YbfA, highlights YbfA as a new antibiofilm target appropriate for small-molecule inhibitor development. Additionally, considering tyrosol's presence in foods like olive oil, regular consumption might offer some preventive benefits against bacterial infections, especially those related to biofilm formation.

In conclusion, by understanding the mechanism of biofilm formation with tyrosol, this study elucidates a mechanism by which low levels of NO are regulated to promote biofilm formation during the late stage of its development. YbfA is activated to stimulate *E. coli* biofilm formation via FNR in this late stage under limited oxygen conditions. Notably, tyrosol, a safe food ingredient, inhibits YbfA-mediated anaerobic biofilm formation, thereby remarkably increasing the susceptibility of biofilm cells to antibiotics. Thus, this study reveals a promising strategy for developing therapeutics to combat multidrug-resistant infections caused by biofilm-forming pathogenic bacteria. Furthermore, our finding provides new insights into understanding the complex mechanisms of biofilm development.

## Methods
### Materials
Tyrosol, FC, and C-PTIO potassium salt were purchased from Sigma-Aldrich (St. Louis, MO, USA; #79058, #53796, and #C221, respectively). Complestatin was isolated from *Streptomyces chartreusis* AN1542 mycelia as stated in our previous study[72].

### Bacterial strains
*E. coli* BW25113 and the Keio *E. coli* knockout library were from the National Institute of Genetics (Shizuoka, Japan); *P. aeruginosa* PA14 was provided by Y. H. Cho (Cha University, Seoul, Republic of Korea). *E. coli* BL21(DE3) (Enzynomics, Korea) was used as the host strain for expression of the YbfA and YfeR proteins. *E. coli* was grown in Luria-Bertani (LB) medium (10 g/L tryptone, 5 g/L yeast extract, and 10 g/L NaCl; pH 7.2) at 37 °C and 220 rpm and was supplemented with kanamycin (50 mg/L) as required.

## Biofilm formation assay

Biofilm formation assays were conducted using a 96-well polystyrene microtiter plate. *P. aeruginosa* PA14 and *E. coli* BW25113 cells were cultured overnight and then diluted 100-fold in M63 medium. A volume of 100 µL of the diluted cell suspensions was added to wells of the microplate. Test compounds or DMSO as a solvent control were added to the wells. The microplate was incubated statically for 9 h for *E. coli* and 18 h for *P. aeruginosa* at 37 °C. After determining planktonic cell viability through OD measurement at 600 nm or viable cell counting, unattached cells and media were discarded. The attached biofilms were stained with 120 µL of 0.1% crystal violet for 10 minutes. After washing to remove excess dye, the bound crystal violet was dissolved with 30% aqueous acetic acid for 15 min. The absorbance of the eluted crystal violet at 550 nm was measured using a VersaMax microplate reader (Molecular Devices, California, USA).

## Screening of the fungal fermentation extract library

Screening of the fungal fermentation extract library was conducted to identify extracts that reduce NO production and, consequently, decrease biofilm production by *P. aeruginosa* PA14. Initially, each extract, dissolved in DMSO, was added to the wells in the 96-well biofilm formation assay at a 2% (v/v) concentration. Extracts that exhibited more than a 40% inhibition of biofilm formation were chosen for further screening. Among the selected extracts, those for which biofilm formation returned to normal levels upon treatment with C-PTIO were ultimately selected for the investigation of active compounds.

## Confocal laser scanning microscopy for biofilm visualization and intracellular NO detection

Confocal laser scanning microscopy for biofilm visualization[73] and intracellular NO detection[74] was carried out based on a previously described protocol with some modifications. *E. coli* biofilms were cultivated on 15-mm$^2$ glass coverslips (Matsunami Glass Ind., Ltd. Japan; #0111550). These sterile coverslips were positioned vertically within 24-well plates. *E. coli* bacteria were statically incubated at 37 °C for 24 h. Subsequently, coverslips underwent two rounds of washing with sterile phosphate-buffered saline (PBS). The biofilms on the coverslips were then stained using SYTO9/propidium iodide, following the instructions of the LIVE/DEAD BacLight Bacterial Viability Kit (Invitrogen Molecular Probes, USA; #L13152). After staining, the biofilms underwent another round of washing with sterile PBS to eliminate planktonic bacteria and dyes. The biofilms were then visualized using CLSM (Carl Zeiss LSM800, Jena, Germany) by exciting the biofilm samples at 490 nm (emission: 635 nm) and 480 nm (emission: 515 nm). As for the detection of intracellular NO, the biofilms on the coverslips were stained with the fluorescent NO probe DAF-2DA (Sigma; #D2813) at a concentration of 20 µM for 1 h. After washing the biofilms with sterile PBS, the detection of NO was carried out by exciting the biofilms at 480 nm (emission: 515 nm).

## Antibiotic susceptibility assay

The antibiotic susceptibility of planktonic and biofilm cells of *E. coli* was assessed in a 96-well polystyrene microtiter plate. Planktonic and biofilm cells from the same growth stage were used. For the planktonic cell culture, the cells were obtained after 14 h of cultivation in M63 medium. The media were then replaced with fresh medium containing antibiotics and dispensed in a 96-well polystyrene microplate. After 1 h of incubation, viable cells were counted. For the biofilm cells, the biofilms were cultivated for 12 h either in the absence or presence of tyrosol, with FC and complestatin as controls, in a 96-well polystyrene microplate. Subsequently, the medium was replaced with fresh medium containing antibiotics following 1 h of incubation and subsequent viable cell counting.

## Viable cell counting

Overnight cultures of *E. coli* and *P. aeruginosa* were diluted 100-fold in M63 medium using Falcon tubes. Cell suspensions (100 µL) were then distributed into the wells of a 96-well polystyrene microplate. Biofilms were allowed to grow under static conditions for 9 h for *E. coli* and 18 h for *P. aeruginosa* at 37 °C in the presence of varying concentrations of test compounds. To obtain planktonic cells, unattached cells were collected by centrifugation at $12,000 \times g$ at 4 °C for 10 min. For disengaging bacterial cells from the biofilms, sonicating for 2 min at a frequency of 40 kHz with a 50 W power output was performed. Viable cells from both planktonic and biofilm cultures were enumerated through serial dilution and plating.

## Quantitative analysis of EPS in biofilms

To evaluate EPS in *E. coli* and *P. aeruginosa* biofilms, a previously described method was employed with some modifications[75]. Both bacterial strains were incubated in 96-well plates with M63 medium supplemented with test compounds for 9 h for *E. coli* and 18 h for *P. aeruginosa* at 37 °C. Subsequently, the medium was collected and then combined with the residue obtained by washing the inner well surfaces with PBS to gather loosely associated EPS. The combined medium was utilized to extract the free EPS, while the biofilms that remained attached were used for the extraction of bound EPS. For the extraction of free EPS, the combined medium was centrifuged at $3000 \times g$ for 15 min at 4 °C and then the supernatant was centrifuged again at $10,000 \times g$ for 30 min at 4 °C to eliminate any remaining cells. The resulting supernatant, containing free EPS, was subsequently precipitated using a 1:3 volume of ethanol and stored at -20 °C for 18 h. Free EPS was then separated by centrifugation at $10,000 \times g$ for 15 min at 4 °C. The extract was resuspended in ultrapure water and dialyzed against ultrapure water to remove ethanol using Spectra/Por® 7 MWCO 2000 (Carl Roth; #E857.1). For the extraction of bound EPS, the attached biofilms were gently sonicated to release EPS from the cells using the same sonication conditions as employed for the viable cell counting in biofilm cells, and then detached cells and medium were collected and centrifuged at $3000 \times g$ for 15 min at 4 °C. EPS was not detectable in the resulting cell pellet using the EDTA method[76], which indicates the complete detachment of EPS. Thus, bound EPS was extracted from the resulting supernatant in the same manner as described above. Both bound and free EPS were stored at -20 °C until needed for further analysis. The carbohydrate content was determined using the Anthrone method[77] with glucose as the standard. The protein content was measured using a Pierce BCA protein assay kit (Thermo Scientific, Waltham, MA, USA; #23225), while the DNA content was measured using the propidium iodide assay.

## Quantitative cellular c-di-GMP analysis by LC–MS/MS

Overnight cultures of both bacteria were diluted 100-fold in M63 medium. The diluted cultures (100 µL) were dispensed into wells of a 96-well plate. Different concentrations of test compounds or DMSO, serving as a solvent control, were added to the cultures. *E. coli* and *P. aeruginosa* cells were then statically incubated at 37 °C for 18 h and 9 h, respectively, in the presence of test compounds. After removing unattached cells and medium, the surface-attached biofilm cells were washed with distilled water. To extract c-di-GMP concentrations from biofilm cells, 100 µL of M63 medium was added to each well. Biofilm cells were detached from the wells using sonication for 2 min, and the cells were collected. The collected cells were centrifuged at $12,000 \times g$ for 10 min at 4 °C, and the cells were harvested. The cells were subsequently resuspended in M63 medium. To this suspension, perchloric acid was introduced at 70% v/v, achieving a final concentration of 0.6 M. Initially, these biofilm cell mixtures containing perchloric acid were allowed to incubate on ice for a duration of 30 min, and then the remaining experiment were carried out at 4 °C. The suspensions were then subjected to centrifugation for 10 min at $12,000 \times g$, maintaining a

temperature of 4 °C. Supernatants from the cell suspensions were collected in 1.5-mL Eppendorf tubes and assessed for protein concentration using the Pierce BCA protein assay kit. To neutralize the pH of nucleotide extracts, 20 μL of potassium bicarbonate (2.5 M) solution was added, followed by brief centrifugation to collect the samples. Nucleotide extract supernatants were transferred to fresh tubes and centrifuged again to remove perchlorate salt precipitates. The c-di-GMP concentrations in the supernatants were quantified using LC-MS/MS. The samples were analyzed using an HPLC system (Luna C18(2), 100 × 2.0 mm, 3 μm, Phenomenex, Torrance, CA, USA) connected to a QTrap 3200 with a Turbolon Spray source (AB SCIEX, Singapore).

### PDE and DGC activity assays
Overnight cultures of *E. coli* and *P. aeruginosa* were diluted 100-fold using LB medium as the diluent. Different concentrations of test compounds were introduced into the diluted cultures. Subsequently, *E. coli* and *P. aeruginosa* cultures were incubated for 6 h and 24 h, respectively, at 37 °C and 220 rpm in a shaking incubator. After incubation, the cultures were centrifuged at 12,000 × *g* for 10 min at 4 °C. The harvested cells were resuspended in 1 mL of LB medium. DGC and PDE activities were evaluated using two separate buffers (200 μL each) added to the harvested cell suspensions. For the PDE activity assay, the buffer composition included 50 mM Tris base (pH 8.1), 50 mM sodium chloride, 1 mM manganese chloride, and 5 mM bis-para-nitrophenyl phosphate (pNPP) – a commonly used artificial substrate for c-di-GMP-specific PDE. For the DGC activity assay, the buffer consisted of 250 mM sodium chloride, 75 mM Tris-hydrochloride (pH 7.8), 25 mM potassium chloride, and 10 mM magnesium sulfate. The cell suspensions were lysed through sonicating 10 times for 30 sec each. The lysed cell suspensions were then subjected to a 2 h incubation at 37 °C and 60 rpm for the PDE reaction. For the PDE reaction, p-Nitrophenol formation from bis-pNPP was quantified using an ELISA reader at a wavelength of 410 nm. In the DGC assay, 25 micromoles of GTP as the substrate were added to the supernatants for a 2 h incubation under similar conditions. Cell lysates without the addition of GTP were treated as blanks. The concentration of c-di-GMP, synthesized via the interaction of DGC enzyme with GTP, was quantified using LC-MS/MS. Total protein concentration was determined using Pierce BCA protein assays and used for the normalization of PDE (OD at 410 nm) and DGC (c-di-GMP in pmol) activities.

### Screening the *E. coli* Keio collection library
*E. coli* mutants with phenotypes similar to tyrosol-treated cells were identified from the Keio mutant library. The Keio mutant library, representing 3801 genes, was cultivated overnight in 96-well plates with LB medium supplemented with 15 μg/mL kanamycin. Subsequently, 3 μL of the overnight culture was added to 100 μL of M63 medium in 96-well plates and incubated for 18 h at 37 °C. After determining planktonic cell viability by measuring OD at 600 nm or viable cell counting, biofilm assays were conducted. Primary hits showing at least 30% less biofilm formation compared to the parent strain *E. coli* BW25113 from the Keio library were subjected to three rounds of rescreening. Biofilm-forming ability of primary hits was further assessed in the presence of C-PTIO, where hits displaying enhanced biofilm production were chosen as secondary hits. For the secondary hits, intracellular c-di-GMP levels and PDE activity were analyzed. Hits exhibiting lower c-di-GMP levels and higher PDE activity than BW25113 were selected for subsequent overexpression assays.

### Overexpression assay
Four *E. coli* overexpression strains were generated by amplifying the wild-type *napA*, *ybfA*, *yfeR*, and *fnr* genes, both with and without their respective promoter regions (-1 to -150 bp), from *E. coli* W3110 genomic DNA using PCR with the specified primers (see Table S2 for primer details). The resulting PCR products were then cloned into the pBAD-

TOPO TA expression vector (Invitrogen, Carlsbad, CA, USA; #K430001), creating recombinant pBAD plasmids that enable gene expression under the control of the arabinose promoter[78]. Subsequently, each of these pBAD recombinant plasmids was introduced into *E. coli* BW25113 through the process of electroporation, leading to the generation of the respective overexpression strains. In the case of the *napA* gene along with its promoter region, the recombinant pBAD-p*napA* plasmid was introduced into both *E. coli* BW25113 and the *E. coli* *napA* mutant, resulting in the creation of *E. coli* BW25113 (pBAD-p*napA*) and *E. coli* ΔnapA (pBAD-p*napA*), respectively.

### Cloning, expression, and purification of YbfA and YfeR
The *ybfA* and *yfeR* genes were cloned from *E. coli* W3110 genomic DNA using PCR with the specified primers (see Table S3 for primer details). The amplified PCR products were purified, digested with *Nde*I and *Hind*III, and subsequently cloned into the corresponding sites of the pET28a vector with a 6xHis tag. The resulting recombinant expression vectors were then transformed into *E. coli* BL21(DE3). The *E. coli* BL21(DE3) culture, transformed with the recombinant expression vector, was induced with isopropyl-1-thio-β-D-galactopyranoside (IPTG) to a final concentration of 0.5 mM. After centrifugation of the cell lysate, recombinant YbfA and YfeR proteins were purified using nickel-nitrilotriacetic acid (Ni-NTA) column chromatography (Qiagen; #30210). The fractions eluted with 100 mM Tris-HCl (pH 8.0) buffer containing 50 and 100 mM imidazole were pooled and concentrated with an Amicon Ultra−15 instrument (Millipore; #UFC900324). The concentrated enzyme was dialyzed overnight at 4 °C against 100 mM Tris-HCl (pH 7.5) using dialysis membranes (Viskase; #44311) with a 3.5 kDa cut-off and stored at 4 °C until use. The purified proteins were confirmed by 10% (w/v) SDS-PAGE, and their concentrations were determined using a UV spectrometer.

### Molecular docking
The 3D structure of the YbfA protein was predicted using ColabFold (AlphaFold2 integrated with MMseqs2 for Many-against-Many sequence searching)[48] and the model with the highest pLDDT (predicted Local Distance Difference Test) value was selected. The ligand tyrosol was generated using ChemDraw 16.0.1.4, and energy-minimized with Open Babel 3.1.1 (http://openbabel.org). This prepared ligand tyrosol was then docked into the predicted model of YbfA protein using Autodock Vina 1.2.3[49]. During docking, the docking grid box size was set to encompass the entire protein, but rotatable bonds in the ligand were not set for flexible docking. Among the docking poses obtained, one with low binding energy and favorable orientation was chosen for further analysis. The docking poses were visualized and analyzed for interactions between the protein residues and the ligand by PyMOL 2.5.5 Molecular Graphics System (https://pymol.org/2/) and LigPlot+ v.2.2[79].

### Generation of *ybfA* gene and YbfA protein variants
To generate *ybfA* variants in the pBAD-*ybfA* plasmid for in vivo binding assays of YbfA variants to tyrosol, site-directed mutagenesis was conducted using the Muta-direct™ Site-directed Mutagenesis Kit (Intron biotechnology; #15071). The PCR mixtures, with a total volume of 50 μL, included 2 μL of plasmid, 1 μL of forward primer, 1 μL of reverse primer, 5 μL of reaction buffer, 2 μL of dNTP mixture, and 1 μL of Muta-direct™ enzyme (see Table S4 for primer details). After PCR amplification, the parental plasmid was digested with Mutazyme™ to eliminate any unmutated template DNA. The resulting amplified products were then transformed into *E. coli* DH5α. Mutations within the plasmids were confirmed by sequence analysis. Subsequently, each mutated plasmid was transformed into either *E. coli* BW25113 to construct *ybfA* variant-overexpressing strains or the Δ*ybfA* strain to construct Δ*ybfA* strains complemented with the *ybfA* variant. For preparing YbfA variant proteins for ITC experiments, genes encoding

each *ybfA* variant were cloned from the pBAD plasmid, containing the respective *ybfA* variants, into the pET28a vector using PCR with the same primers used for the *ybfA* cloning. Following sequence verification, the YbfA variants were expressed and purified according to the aforementioned methods.

## Isothermal calorimetry

ITC experiments were conducted using a MicroCal VP-ITC calorimeter (Malvern Instruments Ltd, Worcestershire, UK) at 25 °C. The YbfA protein was diluted to a concentration of 6 µM in a buffer containing 100 mM Tris, 10 mM KCl, pH 7.5. Tyrosol was prepared in the same buffer at a concentration of 100 mM. Thirty consecutive 5 µL injections of tyrosol were titrated into 1 mL of YbfA in the sample cell, with a 150 s interval between injections. Nonlinear curve fitting was performed using a single-site binding model with Origin 7.0 software. The ITC titration was repeated twice.

## Electrophoretic mobility shift assay

EMSA was conducted using a Lightshift Chemiluminescent EMSA Kit (Thermo Fisher Scientific, Waltham, USA; #20148). Oligonucleotide probes were synthesized and labeled at the 5′-end with biotin (Macrogen, Seoul, Korea). DNA binding reactions were performed in a 10 µL system containing varying amounts of purified YbfA or YfeR recombinant proteins, 10 fmol of biotin-labeled 150-bp *yfeR* promoter DNA (P*yfeR*) or 150-bp *fnr* prompter DNA (P*fnr*) probes, 1 µL binding buffer, and 0.5 µL poly (dI·dC). For competition assays, increasing excesses of unlabeled probes were added to the reactions. The mixture was incubated at 25 °C for 30 min and then electrophoresed at 100 V for 1 h on 6% (w/w) polyacrylamide gels. After being transferred to a positively charged nylon membrane (Biodyne B Nylon Membranes, Thermo Fisher Scientific, Waltham, USA; #77016), the membranes were cross-linked under an ultraviolet lamp. The membrane was treated with the Lightshift Chemiluminescent EMSA Kit and then imaged using a ChemiDoc Imaging System (Bio-Rad).

## Expression and RT–qPCR assay of denitrification-associated genes and related genes

*E. coli* and *P. aeruginosa* culture samples for RT–qPCR analysis were prepared in a similar manner as for c-di-GMP analysis, although the incubation time for *P. aeruginosa* biofilm culture was extended to 12 h. The resulting cultures were subsequently subjected to centrifugation at 12,000 × *g* and 4 °C for 10 min. Total RNA was isolated utilizing TRIzol reagent (Invitrogen; #15596026), following the manufacturer's protocol. To facilitate RT–qPCR analysis of target gene expression, cDNA was synthesized, and RT–qPCR was conducted using a Bio-Rad CFX-96 real-time system (Bio-Rad, Hercules, CA, USA) with the primers listed in Table S1. The expression of mRNA was normalized employing the endogenous rpoD gene as a reference for normalization.

## Differential RNA-seq and data analysis

For the differential RNA-seq analysis, the Δ*wzb*, Δ*nirC*, Δ*ybfA*, and wild-type *E. coli* culture samples were prepared following a similar procedure to the RT–qPCR analysis. The cultures underwent centrifugation at 12,000 × *g* and 4 °C for 10 min. Total RNA was extracted using TRIzol reagent (Invitrogen), adhering to the manufacturer's instructions. Quantification of total RNA concentration was achieved using Quant-IT RiboGreen (Invitrogen, #R11490). To ascertain the integrity of the total RNA, samples were subjected to analysis on a TapeStation RNA ScreenTape (Agilent). Only RNA samples of high quality with a RIN exceeding 7.0 were employed for RNA library construction. Library preparation was performed independently for each sample, utilizing 1 µg of total RNA and an Illumina TruSeq Stranded mRNA Sample Prep Kit (Illumina, Inc., San Diego, CA, USA, #RS-122-2101). Initially, bacterial rRNA was depleted using the NEBNext rRNA Depletion Kit (Bacteria) from NEB. Subsequently, the remaining RNA was fragmented into smaller fragments through the use of divalent cations under increased temperature. These fragmented RNA segments were then converted into first-strand cDNA through reverse transcription employing SuperScript II reverse transcriptase (Invitrogen; #18064014) and random primers. Following this, second-strand cDNA synthesis was performed using DNA Polymerase I, RNase H, and dUTP. The resulting cDNA fragments underwent end repair, the addition of a single 'A' base, and adapter ligation. The ensuing products were purified and amplified via PCR to produce the final cDNA library. Library quantification was carried out using KAPA Library Quantification Kits for the Illumina Sequencing platform, following the qPCR Quantification Protocol Guide (KAPA). Library quality was verified using a TapeStation D1000 ScreenTape (Agilent). Indexed libraries were subsequently subjected to paired-end (2 × 100 bp) sequencing on an Illumina NovaSeq (Illumina, Inc., San Diego, CA, USA) platform, conducted by Macrogen Incorporated. (i) **Data processing and analysis**. Sequencing reads were obtained through paired-end sequencing using the Illumina NovaSeq platform. Prior to analysis, the Trimmomatic v0.38 tool was employed to eliminate adapter sequences and trim low-quality bases. The resulting cleaned reads were aligned to the *E. coli* BW25113 (ASM1681185v1) genome utilizing Bowtie 1.1.2[80,81]. The reference genome sequence and gene annotation data were obtained from the NCBI Genome assembly and NCBI RefSeq database, respectively. Following alignment, HTSeq v0.10.0 was used to assemble the aligned reads into transcripts and to estimate their abundance[82]. Quantification was performed at both the gene and transcript levels, presenting raw read counts and FPKM (fragments per kilobase of transcript per million mapped reads) as metrics. (ii) **Differential gene expression analysis**. Statistical analyzes of differential gene expression was conducted using DESeq2 v1.24.0[83] with raw counts as input. In the quality control step, genes exhibiting nonzero counts across all samples were selected. Principal component analysis (PCA) and multidimensional scaling (MDS) plots were generated to confirm expression similarities among samples. Relative log expression (RLE) normalization was applied to the filtered dataset to mitigate library size variations across samples. The statistical significance of the differential expression of genes was determined using the nbinomWaldTest function in DESeq2. Fold change and *p*-values were extracted from the nbinomWaldTest results. The list of significantly differentially expressed genes was refined by applying criteria |fold change| ≥2 and raw *p*-value < 0.05.

## Statistics and reproducibility

Statistical tests, the number of events quantified, the standard deviation of the mean, and statistical significance are reported in figure legends. The two-sided Student's *t* test was used to analyze the data (Excel software, Microsoft, Redmond, WA, USA). No statistical method was used to predetermine sample size. No data were excluded from the analyzes; The experiments were not randomized; The Investigators were not blinded to allocation during experiments and outcome assessment.

## Reporting summary

Further information on research design is available in the Nature Portfolio Reporting Summary linked to this article.

## Data availability

The RNA-seq raw data have been deposited in the NCBI Sequence Read Archive (SRA) (https://www.ncbi.nlm.nih.gov/sra) with the code BioProject accession PRJNA1063457. All other relevant data are available in its Supplementary Information files. Source data are provided with this paper.

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

## Acknowledgements

This work was supported by the Basic Science Research Program through the National Research Foundation of Korea (NRF) funded by the Ministry of Education (grant number NRF-2021R1I1A2048905), the Ministry of Science and ICT (MSIT) (2023M3A9H5061759), and the Korea Research Institute of Bioscience & Biotechnology (KRIBB) Research Initiative Program, Republic of Korea.

## Author contributions

W.G.K. conceived and designed the study. H.Y.C. performed the experiments. W.G.K. and H.Y.C. analyzed the experimental data. W.G.K. wrote the manuscript.

## Competing interests

The authors declare no competing interests.
