## [Peer Review File · Nature Communications]

REVIEWER COMMENTS

Reviewer #1 (Remarks to the Author):

The work titled “Tyrosol blocks *E. coli* anaerobic biofilm formation via YbfA and FNR, increasing biofilm cell susceptibility to antibiotics” elucidates the mechanism of actions of naturally occurring compound Tyrosol. The study describes several systematic experiments involving mutant strains, complementation, and overexpression that elucidate the mechanism of action and effectiveness of the test compound. Overall, the work is significant and well performed, although it was difficult to follow at times.

The work is significant and important. The manuscript unveils a novel mechanism that regulates biofilm formation by influencing NO homeostasis. However, the approach and problem definition used in this study is somewhat comparable to previous work by the authors (doi: 10.1128/mBio.00878-20) and relies on targets that upregulate NO levels. Overall, the data and observations derived based on sufficiently described assays substantiate the outcomes of the study. However, a mechanism of biofilm prevention may not translate to mechanism of dispersal of mature biofilms and requires further investigation.

Major comments:

1. The introduction starts out: “To discover fungal fermentation extracts that stimulate NO production, thereby reducing biofilm production by *P. aeruginosa* PA14, we screened 2,056 extracts. This process led us to select extracts from two specific fungal strains. However, only one strain's fermentation extract inhibited biofilm formation by prompting NO production, without affecting cell viability...” There appears to be no details on this screening in the experimental or results sections, and Fig 1A really only shows the active material tyrosol. Was the screening part of a prior study (no citations)? Citations and/or more details need to be provided.

2. There is a repeated mention that “Mature biofilms are reportedly resistant to antimicrobials due to oxygen limitation within their interior.” When looking up biofilms resistance to antibiotics, common reasons listed in many sources include limited diffusion of antimicrobial to the cells, levels of metabolic activity inside the biofilm. Oxygen is not really mentioned. While two citations are listed to support the oxygen limitation within their interior, one (ref 6) does not really state this strongly, but instead states: based on the experimental data, it is likely that the observed resistance of biofilms can be attributed to formation of ampicillin-resistant subpopulations in the deeper layers of mature biofilms but not in young colony biofilms.” The other does mention oxygen levels. Thus, the central part is not that well supported by prior literature and perhaps the wording should be changed. They go further and state: Line 46-47/502-503: While it has been shown that anoxic zone of mature biofilms corresponds with drug resistance, it is not clear how authors arrived at the estimate “Oxygen depletion reportedly accounts for at least 70% of the antibiotic resistance observed in mature biofilm cells of *P. aeruginosa*” based on reference 6 and 9. Please include additional reference supporting this claim.

Minor Comments:

1 Line 102: Missing punctuation.

2 Line 129-130: Fungal product tyrosol and its anti-biofilm activity has been previously reported. Authors are requested to cite previous work supporting such observation. For example,

<https://doi.org/10.1155/2015/456463>, <https://doi.org/10.1111/jam.13070>. Also, Tyrosol may exert other pharmacological effects. How will tyrosol and other ligands with similar structure and activity influence choice in proposed strategy for the treatment of biofilm using YbfA as a target?

3 Line 458-460, How does tyrosol influence the effectiveness of existing antibiotics on the already developed/mature biofilms? The efficacy of tyrosol has been shown through reduction in adherent cells and biofilm matrix amount. However, it is not clear how effective the tyrosol or chemicals with similar mechanism of action will be with the mature biofilms given the fact that mature *E. coli* biofilms are reported to exhibit increased antibiotic resistance. The study exposes bacterium to the active compound before the formation of biofilms (Line 558-559).

4 Line 547-548, Interference of biofilm in OD600 readings: It is not clear how the authors accounted for any interference arising out of surface attached biofilms during the measurement of CFU using OD600.

5 Lack of raw data from replicates experiments: In absence of data (except as represented in the graphs) validation of standard deviations and statistical significance is difficult. Availability of measurement values from replicate experiments will be helpful. For example, figure 2 (A) (i) shows statistical significance in the cases of BW25113 for measurements upwards 0.3 (x-axis) using #, ##, ###, ### representing p-values <0.01, <0.001, and <0.0001. It is difficult to ascertain how the values were estimated. Consider adding further description in the figure legends or include information within plots that is readily accessible to identify compared categories/groups.

6 The supplementary excel sheet name should be updated to English text for consistency.

Reviewer #2 (Remarks to the Author):

Reviewer #3 (Remarks to the Author):

The authors present in their manuscript the natural product tyrosol as inhibitor of biofilm production in *E. coli*. Interestingly, tyrosol acts as an enhancer of NO production. YbfA was identified as primary target of tyrosol. Subsequently two regulators, namely YfeR and FNR, were shown to be involved downstream of ybfA in the modulation of biofilm formation. FNR the regulator for denitrification served as the link to nitric oxide production. The nitrate reductase gene napA was found to be induced upon tyrosol treatment. By deletion and complementation analyses, as well as RT-PCR and RNAseq the role of these three genes ybfA, yfeR and napA was elucidated.

The outstanding feature of the work is the use of tyrosol as antibiofilm agent and to understand the effect of endogenous nitric oxide-production on biofilm production. The identification of targets of tyrosol is of interest to the scientific community, as it represents an important step in antibiofilm strategies. The authors present the regulatory cascade that finally leads to upregulation of the nitrate reductase gene napA and thereby interlinks the tyrosol treatment with host denitrification metabolism. Furthermore, the antibiofilm effect of tyrosol was combined with antibiotic susceptibility screens with promising results.

The novelty and importance of the research for the scientific community is good. Several papers have been published showing the antibiofilm activity of different natural compounds thereby providing new targets for further medical research (Rojita et al. 2020 *Frontiers in Microbiology*, 11, <https://doi.org/10.3389/fmicb.2020.566325>).

Originality and significance: In this work a combination of phenotypic and transcriptional analyses led to the identification of a new strand in the regulatory network of biofilm formation consisting of YbfA, YfeR and FNR. A very similar approach was reported previously to identify the nitrite transporter as target of a bioactive compound by this group (Park et al. 2020 *mBio*. 11(4):e00878-20. doi: 10.1128/mBio.00878-20).

Data and methodology: The study is very comprehensive on transcriptional data. Relative gene expression levels are shown. The explanation which reference conditions / genes were used is not always clearly stated. Quality of data is good, with remarkable high reproducibility along all experiments. Statistics is included and sound. I would recommend to deposit RNAseq data on a public database. Some experiments and controls are not well explained. Global measurements for cellular PDE or DGC activity is not best representative for biofilm function and should be considered with care (Hengge et al. 2021 *Trends Microbiol.* 29(11):993-1003. doi: 10.1016/j.tim.2021.02.003.).

Clarity and context: The manuscript is very intense and not easy to follow. The last part of the introduction needs a strong rework. Further, the manuscript could be significantly improved including a scheme for denitrification (genes) to explain targets of RT-PCR and an improvement of the proposed regulatory cascade. The construction of complementation and overexpression strains is confusing, and the results aren't conclusive especially in absence or presence of arabinose. The expectations should be presented to understand the results clearer. To me it was not made clear either, if the regulation of yfbA, YfeR and FNR is only on transcriptional level (meaning in the 5'UTR of the genes?).

Conclusions: In the present work, special emphasis was put upon the endogenous NO production. With FNR being the major regulator of denitrification in *E. coli*, the authors established the linkage to nitric oxide formation. FNR regulates a large number of genes, e.g. nir, nrf and nap, involved in denitrification. To my surprise, only napA and fnr were found to be upregulated in this study. Furthermore, the increase of nitric oxide was assumed here from increased napA expression. The nitrate reductase NapA catalyzes the reaction from nitrate to nitrite. For the formation of nitric oxide in denitrification process the nir genes, coding for holo nitrite reductase, are required. These genes were not found to be induced in any of the experimental tests. The nitric oxide levels in the cell haven't been monitored. Genes or proteins involved in NO homeostasis, namely Hmp and NorCB were not further investigated. Solely, the use of C-PTIO in phenotypic assays indicated that nitric oxide is some way involved. In my humble opinion, there is a lack of conclusive arguments for the assumption that tyrosol / YbfA influence biofilm formation through endogenous nitric oxide formation.

Suggested improvements:

Line 104: please phrase better: the referred biofilm inhibitors do not produce NO

Line 106: identification of antibiofilm activity of tyrosol (karkovich et al. 2019): wrong reference

Line 240: Essential information is missing on the features of YbfA

Line 408: controls are not clear, why nirB transcription in nirC mutant? What should that look like?

line 419: why should tyrosol treated cells show reduced susceptibility to antibiotics?

The authors should refer in the discussion also to the biofilm related functions of tyrosol in

Pseudomonas aeruginosa as previously published (QS inhibitor, antioxidant). [Chang et al. 2019]. Include in the discussion that in a former publication YbfA is related to another regulatory system (BasS/BasR) [Chen et al. 2021]. How would the authors integrate these data with their findings? How do these results affect the conclusions drawn in the present work?

Furthermore, the authors do not discuss the NO homeostasis in the cell, possible degradation and scavenging (flavo-hemoglobin, denitrification itself, tyrosol as antioxidant, scavenger molecule etc). Which role would play a NO sensor/ regulator (NosP, HNOX) in this regulatory cascade [Hossain & Boon 2017]?

Further references should be included:

Aiping Chang, Shiwei Sun, Li Li, Xiaoyun Dai, Hui Li, Qiaomei He, Hu Zhu (2019) Tyrosol from marine Fungi, a novel Quorum sensing inhibitor against *Chromobacterium violaceum* and *Pseudomonas aeruginosa*. *Bioorganic Chemistry*, Volume 91, 103140. <https://doi.org/10.1016/j.bioorg.2019.103140>.

Hossain S, Boon EM. Discovery of a Novel Nitric Oxide Binding Protein and Nitric-Oxide-Responsive Signaling Pathway in *Pseudomonas aeruginosa*. *ACS Infect Dis*. 2017 Jun 9;3(6):454-461. doi: 10.1021/acsinfecdis.7b00027.

Final comments:

Overall, the manuscript contains a very extensive dataset with interesting data. Nevertheless, clarity in the presentation and interpretation of data must be significantly improved. The manuscript needs major revision for publication. Apart from this, I do not want to judge whether the significance of the results is sufficient for publication in *Nature Communications*.

Reviewer #4 (Remarks to the Author):

In this manuscript, the authors screened biofilm-dispersing compounds and identified a compound tyrosol that is able to inhibit *E. coli* and *P. aeruginosa* biofilm formation by enhancing nitric oxide (NO) production. Next, they investigated genetic regulation involved in the tyrosol-mediated biofilm dispersal and identified that YbfA, a previously unknown function protein, is the target of tyrosol. Using RNA-seq based transcriptomic analysis, the authors revealed that YfeR, the downstream protein of YbfA, regulates anaerobic transcriptional regulator FNR and further affect NO production and c-di-GMP signalling. In general, the authors discovered a potentially interesting biofilm dispersal compound, but their finding is too preliminary, and the genetic characterization of its working mechanism is not very solid and lack of in depth characterization. Here are my major points:

Methodology part:

- 1) Biofilm assay was performed only on microplates, which is usually unstable and depending on the handling by persons and even microplate brands can affect the results. Normally, flow cell biofilm experiment is required together with the microplate static biofilm assay to make a solid observation of biofilm phenotypes;
- 2) The biofilm extracellular polymeric substance (EPS) analysis is not well down as well. The staining

approaches they used are not very strict and specific. If using the extraction methods, the authors should use gentle sonication approaches to carefully lose EPS from cells and then started to precipitant polysaccharide, eDNA and proteins for quantification as described in this paper:

<https://pubs.acs.org/doi/10.1021/bm701043c>. If using the in situ staining approaches, the authors should use very specific dyes such as described in this paper:

https://link.springer.com/protocol/10.1007/978-1-4939-0467-9_4;

3) RNA-seq analysis should provide raw data accession number from NCBI (Sequence Read Archive (SRA)); The fold change in the differential gene expression analysis is absolute value or log transformed value?

Conclusion part:

1) The authors provided a series of regulatory mechanism from YbfA to FNR to denitrification pathways which might be affected by tyrosol. However, only RNA-seq and RT-PCR experiments were performed to draw these conclusions. But the fold changes from these analyses are very small for many cases (Figure 1D, Figure 4, Figure 6);

2) There is almost no direct binding or interacting assay (e.g. isothermal titration calorimetry (ITC) assay) to validate that tyrosol binds with YbfA. What domains do YbfA contain and which one can be affected by tyrosol? How it works as a regulator to YfeR? How YfeR regulates FNR. If it is at the transcriptional, the authors should provide electrophoretic mobility shift assay (EMSA) data to show YfeR binds to which region of the promoter of FNR;

3) There is a lack of general investigation about their findings. Only very few strains were tested, which can not show tyrosol is a general biofilm inhibitor for *E. coli* and *P. aeruginosa*.

The modified parts were highlighted in red color in the revised manuscript.

1. Modifications according to the comment of reviewer # 1:

- 1) Citations and/or more details on the screening need to be provided.

Response) Lines 656-662: The screening method has been described in detail in the methods section.

- 2) Please include additional reference that supports the statement, ‘Oxygen depletion reportedly accounts for at least 70% of the antibiotic resistance observed in mature biofilm cells of *P. aeruginosa*’.

Response)

Lines 46-47: The incorrect reference 6 has been deleted, reference 9 has been repositioned, and the reference 12, which provides evidence for the statement, has been added. Additional references 10 and 11, which further support the statement, have also been included.

Line 473: The incorrect reference 6 has been deleted, and reference 12, which provides evidence for the statement, has been added instead.

Lines 569-570: The reference 12, providing evidence for the statement, has been added. Additional reference 67, which supports the statement, has also been included.

3) Minor comments

- (1) Line 102: Missing punctuation.

Response) Lines 102: Punctuation has been added.

- (2) Line 19-130: Fungal product tyrosol and its anti-biofilm activity has been previously reported. Authors are requested to cite previous work supporting such observation. For example, <https://doi.org/10.1155/2015/456463>, <https://doi.org/10.1111/jam.13070>.

Response) Lines 617-620; References 71 and 73: The inhibitory activity of tyrosol on the production of virulence factors in *P. aeruginosa*, as well as its antibiofilm activity against *Streptococcus mutans* and *Candida albicans*, has been cited.

- (3) Also, Tyrosol may exert other pharmacological effects. How will tyrosol and other ligands with similar structure and activity influence choice in proposed strategy for the treatment of biofilm using YbfA as a target?

Response) Lines 615-617: Tyrosol is known to have various beneficial effects on human health, by preventing hypertension, atherosclerosis, chronic heart failure, insulin resistance, and ischemia-related stress. Despite these beneficial effects of tyrosol, the identification of tyrosol derivatives with enhanced antibiofilm activity is still an area of ongoing research.

- (4) Line 458-460: How does tyrosol influence the effectiveness of existing antibiotics on the already developed/mature biofilms? The efficacy of tyrosol has been shown through reduction in adherent cells and biofilm matrix amount. However, it is not clear how effective the tyrosol or chemicals with similar mechanism of action will be with the mature biofilms given the fact that mature *E. coli* biofilms are reported to exhibit increased antibiotic resistance. The study exposes bacterium to the active compound before the formation of biofilms (Line 558-559).

Response) Lines 476, 484 and 486: Tyrosol reduced antibiotic resistance of mature biofilms developed in presence of tyrosol, but did not affect biofilms that were already developed. To clarify this, the phrase ‘treated with’ has been modified to ‘developed in the presence of’.

- (5) Line 547-548, Interference of biofilm in OD600 readings: It is not clear how the authors accounted for any interference arising out of surface attached biofilms during the measurement of CFU using OD600.

Response) Biofilms form on the vertical walls of the 96-well plates rather than on the bottom. Therefore, the biofilms on the walls do not significantly affect the transparency at a wavelength of 600 nm. In addition to the OD600 measurement method, a viable cell assay has also been employed to verify the results from the optical density assay (Fig. S1A).

- (6) Lack of raw data from replicates experiments: In absence of data (except as represented in the graphs) validation of standard deviations and statistical significance is difficult. Availability of measurement values from replicate experiments will be helpful. For example, figure 2 (A) (i) shows statistical significance in the cases of BW25113 for measurements upwards 0.3 (x-axis) using #, ##, ###, ### representing p-values <0.01, <0.001, and <0.0001. It is difficult to ascertain how the values have been estimated. Consider adding further description in the figure legends or include information within plots that is readily accessible to identify compared categories/groups.

Response)

In the Data availability section, lines 887: The raw data have been provided in a source data file.

In the legends of Fig. 3: 'untreated BW2511' has been changed to 'non-arabinose-treated BW25113'.

In all Figures: When assessing the statistical significance denoted by '#' and '+', these symbols have been underlined on the bars of the two compared groups within plots for easy identification of the groups being compared.

(5) The supplementary excel sheet name should be updated to English text for consistency.

Response) The excel sheet name has been updated to English text.

2. Modifications according to the comments of reviewer # 3:

1) Data and methodology: Relative gene expression levels are shown. The explanation which reference conditions /genes have been used is not always clearly stated.

Response)

Statistical references have been provided in each figure legend.

Fig. 8 Bi, lines 445-446 and 478-480: The reasons for using the *nirC* mutant and complestatin as controls have been explained in detail.

2) Data and methodology: I would recommend to deposit RNAseq data on a public database. Some experiments and controls are not well explained.

Response) In the Data availability section, lines 887-888: The RNA-seq raw data have been deposited in the NCBI database.

Lines 323 – 325: The controls have been explained in more detail.

3) Data and methodology: Global measurements for cellular PDE or DGC activity is not best representative for biofilm function und should be considered with care.

Response)

Global activity for cellular PDE or DGC has been measured to confirm the reduction in total cellular c-di-GMP levels following tyrosol treatment.

Fig. S3 and lines 172-187: If the reduction in global c-di-GMP levels by tyrosol leads to inhibition of biofilm formation, then global measurements for cellular PDE or DGC activity, which are responsible for c-di-GMP biosynthesis and degradation, respectively, are necessary to confirm the effects of tyrosol on these levels. Therefore, to determine if the reduction in total c-di-GMP levels by tyrosol contributes to its antibiofilm activity, we conducted tests on the biofilm formation of a c-di-GMP-overproducing mutant (Fig. S3). This data demonstrated that tyrosol's antibiofilm activity is indeed due to the reduction in total c-di-GMP levels. Consequently, analyzing the global activity of c-di-GMP synthesis and degradation enzymes DGC and PDE is necessary.

4) Clarity and context: The last part of the introduction needs a strong rework.

Response) The last part of the introduction has been revised for more conciseness and clarity.

5) Clarity and context: the manuscript could be significantly improved including a scheme for denitrification (genes) to explain targets of RT-PCR and an improvement of the proposed regulatory cascade.

Response) Fig. 1F: The scheme for denitrification (genes), which explains targets of RT-PCR, has been added.

6) Clarity and context: The construction of complementation and overexpression strains is confusing, and the results aren't conclusive especially in absence or presence of arabinose. The expectations should be presented to understand the results clearer. To me it has been not made clear either, if the regulation of yfbA, YfeR and FNR is only on transcriptional level (meaning in the 5'UTR of the genes?).

Response)

To verify the transcriptional regulation of YfeR and FNR by YfbA and YfeR, respectively, EMSA experiments were performed.

Fig 6B and lines 380-386: The binding of YbfA to the promoter region of *yfeR* was confirmed by EMSA experiments.

Fig 7A and lines 397-403: The binding of YfeR to the promoter region of *fnr* has been demonstrated by EMSA experiments.

7) Conclusions: FNR regulates a large number of genes, e.g. *nir*, *nrf* and *nap*, involved in denitrification. To my surprise, only *napA* and *fnr* have been found to be upregulated in this study.

Response) Lines 553-558: It is reported that FNR induces the transcription of *nir*, *nrf*, and *nap* in the planktonic growth under anaerobic conditions in *E. coli*. In this study, FNR partially suppresses the transcription of only *napA* to produce low levels of NO in the maturation-I-stage of biofilm cultures, which is under anaerobic conditions, because only *napA* is upregulated in the *fnr* mutant. It is important to note that in the biofilm cultures of the *nirC* mutant, which is under aerobic conditions and serve as a control, only *nirB* is upregulated but *fnr* is not (Fig. 8Bii). These results shows that denitrification genes are differently regulated in the biofilm cultures depending on the stage of biofilm cultures.

8) Conclusions: For the formation of nitric oxide in denitrification process the *nir* genes, coding for holo nitrite reductase, are required. These genes have been not found to be induced in any of the experimental tests.

Response) Lines 583-592: Nitrate reductase NARs (*nap* and *nar*) catalyze the conversion of nitrate to nitrite, a substrate of NO, while nitrite reductases NIR (*nir* and *nrf*) produce NO from nitrite. In this study, all denitrification genes and related genes, including the nitrite reductase-coding genes (*nirB* and *nrfA*), were expressed in the untreated biofilm cells as a control (Fig. 1F). This is consistent with the previously reported requirement of low levels of NO for biofilm formation (Park et al. 2020 mBio. 11(4):e00878-20. doi: 10.1128/mBio.00878-20). The expression of *napA*, along with *norV* and *fnr*, was upregulated in tyrosol treatment or in the *ybfA* mutant, compared to the untreated cells. The essential role of *napA* in tyrosol-induced NO production and subsequent biofilm inhibition has been demonstrated by the *napA* mutant, complemented strain, and overexpression strain (Fig. 2). Thus, this study suggests that the regulation of *napA*, by either *ybfA* mutation or tyrosol treatment, contributes to increased production of NO by elevating the production of nitrite, a substrate of NIR.

9) Conclusions: Genes or proteins involved in NO homeostasis, namely Hmp and NorCB have been not further investigated.

Response)

Figs. 1F and 2B; Lines 200-202, 204, 211, 215, 230-231 and 583-592: Originally, genes encoding NO-detoxifying proteins (*hmp* and *norVW*) in *E. coli* were investigated together with other denitrification genes. Data related to NorV was not included in the original manuscript in order to focus on the target protein. The relevant data have now been included in this revised manuscript.

10) Conclusions: The nitric oxide levels in the cell haven't been monitored. Solely, the use of C-PTIO in phenotypic assays indicated that nitric oxide is some way involved. In my humble opinion, there is a lack of conclusive arguments for the assumption that tyrosol / YbfA influence biofilm formation through endogenous nitric oxide formation.

Response)

Fig. 1C and lines 147-153: To further confirm the role of NO in this study, we measured NO levels in biofilm cells by confocal laser scanning microscopy (CLSM) using the fluorescent NO probe 4,5-diaminofluorescein diacetate (DAF-2DA). We observed a dose-dependent increase in NO levels in *E. coli* cells treated with tyrosol compared to untreated cells.

11) Suggested improvements:

(1) Line 104: please phrase better: the referred biofilm inhibitors do not produce NO

Response) Line 104: 'produce NO' has been changed to 'stimulate NO production'.

(2) Line 106: identification of antibiofilm activity of tyrosol (karkovich et al. 2019): wrong reference

Response) Reference 43, Line 106: The reference has been changed to a more appropriate one.

(3) Line 240: Essential information is missing on the features of YbfA

Response) Line 276: The features of YbfA have been described.

(4) Line 408: controls are not clear, why nirB transcription in nirC mutant? What should that look like?

Response) Fig. 8 Bi and lines 445-446: The *nirC* mutant has the same phenotype as the *ybfA* mutant, except for *nirB* upregulation and unchanged *fnr* expression. Thus, the *nirC* mutant has been used as a negative control.

(5) line 419: why should tyrosol treated cells show reduced susceptibility to antibiotics?

Response) Lines 472-476 and 495-497: Mature biofilms are known to be resistant to antimicrobials due to oxygen limitation within their interior. This study reveals that YbfA and FNR function during the maturation-I stage of biofilm formation, a stage where oxygen is limited inside the biofilms. Consistent with the findings that tyrosol inhibits the maturation-I stage of biofilm development through YbfA and FNR, the expression of genes associated with anaerobic metabolism was decreased following tyrosol treatment (Fig. 8C). Therefore, tyrosol increases the susceptibility of biofilm cells to antibiotics by preventing the development of anaerobic biofilms, targeting YbfA and FNR during the early maturation stage.

12) Others

(1) The authors should refer in the discussion also to the biofilm related functions of tyrosol in *Pseudomonas aeruginosa* as previously published (QS inhibitor, antioxidant). [Chang et al. 2019].

Response) Reference 72 and lines 617-618: The inhibitory effects of tyrosol on QS in *P. aeruginosa* have been mentioned and the relevant reference has been cited.

(2) Include in the discussion that in a former publication YbfA is related to another regulatory system (BasS/BasR) [Chen et al. 2021]. How would the authors integrate these data with their findings? How do these results affect the conclusions drawn in the present work?

Response) Lines 529-534: Chen et al reported that the sensitivity of the *ybfA* mutant of *E. coli* K12 to plantaricin is reduced via the BasS/BasR two-component systems in aerobically grown planktonic cultures. This is different from the findings of this study, which show that the sensitivity of the *ybfA* mutant to ciprofloxacin and tobramycin is enhanced via YfeR and FNR in anaerobic biofilms. This suggests that YbfA affects the sensitivity of *E. coli* to antibiotics differently, depending on the culture conditions or the type of antibiotic used.

- (3) Furthermore, the authors do not discuss the NO homeostasis in the cell, possible degradation and scavenging (flavo-hemoglobin, denitrification itself, tyrosol as antioxidant, scavenger molecule etc).

Response) Lines 593-603: The role of NorV in eliminating excess NO during biofilm formation in this study has been discussed, and the regulation of *norV* by FNR has also been mentioned.

- (4) Which role would play a NO sensor/ regulator (NosP, HNOX) in this regulatory cascade [Hossain & Boon 2017]?

Response)

It has been reported that upon NO sensing, NosP regulates biofilm dispersal in *P. aeruginosa*. However, it is not clear whether NosP regulates c-di-GMP PDE in *P. aeruginosa* [Hossain & Boon 2017].

In my humble opinion, BdlA, a sensing domain of c-di-GMP PDE, might be a potential NO sensor in the tyrosol-induced inhibition of biofilm formation in *P. aeruginosa*. Further research is needed to identify an NO sensor in the tyrosol-induced inhibition of biofilm formation.

- (5) Further references should be included:

Aiping Chang, Shiwei Sun, Li Li, Xiaoyun Dai, Hui Li, Qiaomei He, Hu Zhu (2019) Tyrosol from marine Fungi, a novel Quorum sensing inhibitor against *Chromobacterium violaceum* and *Pseudomonas aeruginosa*. *Bioorganic Chemistry*, Volume 91, 103140

Response) Lines 617-618: The reference has been cited as number 71.

Hossain S, Boon EM. Discovery of a Novel Nitric Oxide Binding Protein and Nitric-Oxide-Responsive Signaling Pathway in *Pseudomonas aeruginosa*. *ACS Infect Dis*. 2017 Jun 9;3(6):454-461.

Response) Line 83: The reference has been cited as number 39.

3. Modifications according to the comments of reviewer # 4:

- 1) Methodology part

- (1) Biofilm assay has been performed only on microplates, which is usually unstable and depending on the handling by persons and even microplate brands can affect the results. Normally, flow cell biofilm experiment is required together with the microplate static biofilm assay to make a solid observation of biofilm phenotypes.

Response) Fig. 1C and lines 140-147: The microplate biofilm assay is both reproducible and faster compared to the flow cell biofilm assay, making it suitable for mechanism studies. However, as the reviewer pointed out, the microplate biofilm assay, which is based on crystal violet staining, has limitations in observing solid biofilm phenotypes. Therefore, we further investigated the effects of tyrosol on biofilm formation using confocal laser scanning microscopy (CLSM) and live/dead cells staining. We measured the thickness of biofilms formed on glass coverslips in 24-well plates in the presence of tyrosol. A dose-dependent reduction in biofilm depth by tyrosol treatment was observed, showing a similar inhibition pattern to those obtained in the crystal violet assay in the 96-well plates.

- (2) The biofilm extracellular polymeric substance (EPS) analysis is not well done as well. The staining approaches they used are not very strict and specific. If using the extraction methods, the authors should use gentle sonication approaches to carefully lose EPS from cells and then started to precipitate polysaccharide, eDNA and proteins for quantification as described in this paper: <https://pubs.acs.org/doi/10.1021/bm701043c>.

Response) Fig. S2 and lines 156-110: As the reviewer suggested, bound EPS was extracted using gentle sonication from biofilms cells and subsequently precipitated using the methods described in the paper. Polysaccharides, proteins and eDNA in the extracts were assayed following the methods in the paper. Additionally, free EPS was collected from the medium in the biofilm cultures and analyzed using the same manner.

- (3) RNA-seq analysis should provide raw data accession number from NCBI (Sequence Read Archive (SRA)); The fold change in the differential gene expression analysis is absolute value or log transformed value?

Response) In the Data Availability section, lines 887-888: The RNA-seq raw data have been deposited at the NCBI.

Lines 871-878: For the fold change analysis, first, log₂ fold changes and p-values were extracted using the DESeq2 package from three biological replicates per group. Then,

fold changes were calculated from the log₂ fold changes. Thus, the fold change in the differential gene expression analysis represents an absolute value.

2. Conclusion part

1) The authors provided a series of regulatory mechanism from YbfA to FNR to denitrification pathways which might be affected by tyrosol. However, only RNA-seq and RT-PCR experiments have been performed to draw these conclusions. But the fold changes from these analyses are very small for many cases (Figure 1D, Figure 4, Figure 6);

Response) Although the fold changes of genes differentially expressed in the *ybfA* mutant in RNA seq are small compared to those in the wild-type strain, these fold changes have been statistically significant with *p*-values less than 0.05 (Table S4). We validated the changes using RT-PCR, given the small fold changes in RNA seq (Fig. 4Aii and 5Bii). Furthermore, the differences were also found to be statistically significant in the complemented and overexpression strains (Fig. 6A and 7Bi). Finally, to verify the regulatory mechanism from YbfA to FNR via YfeR, which emerged from the RNA seq and RT-PCR experiments, EMSA experiments were further conducted (Fig. 6B and 7A), as explained in the following sections.

2) There is almost no direct binding or interacting assay (e.g. isothermal titration calorimetry (ITC) assay) to validate that tyrosol binds with YbfA. What domains do YbfA contain and which one can be affected by tyrosol? How it works as a regulator to YfeR? How YfeR regulates FNR. If it is at the transcriptional, the authors should provide electrophoretic mobility shift assay (EMSA) data to show YfeR binds to which region of the promoter of FNR;

Response)

Fig. 3B and lines 271-273: The direct binding of tyrosol to YbfA was demonstrated using the ITC assay.

Figs. 6B and 7A; lines 380-386 and 397-403: The binding of YbfA to the 150-bp upstream region in the *yfeR* promoter and YfeR to the 150-bp upstream region in the *fnr* promoter has been confirmed using EMSA assays. Specific binding of YbfA and YfeR to their respective regions has also been confirmed (Fig. S10).

3) There is a lack of general investigation about their findings. Only very few strains have been tested, which can not show tyrosol is a general biofilm inhibitor for *E. coli* and *P. aeruginosa*.

Response) Figs. S11 and S12; lines 492-494: The ability of tyrosol to enhance the susceptibility of biofilm cells to antibiotics has been demonstrated in several clinical isolates of both *E. coli* and *P. aeruginosa*.

REVIEWER COMMENTS

Reviewer #3 (Remarks to the Author):

I would like to thank the authors for their thorough work. The authors have addressed the raised concerns very thoroughly and carefully. They have added further experiments and results to adequately answer the questions raised in the first review. The manuscript has become significantly more informative and the revision has led to a clear presentation of the findings.

Reviewer #4 (Remarks to the Author):

While reviewing the revised version of this manuscript, I have these general opinion that this manuscript is lack of novelty. Tyrosol is not a very new compound and there are already papers reported it is a quorum sensing inhibitor and biofilm suppressor.

I) 2015 BioMed Research International paper :Effect of Tyrosol and Farnesol on Virulence and Antibiotic Resistance of Clinical Isolates of *Pseudomonas aeruginosa*".

II) 2019 Bioorganic Chemistry paper :Tyrosol from marine Fungi, a novel Quorum sensing inhibitor against *Chromobacterium violaceum* and *Pseudomonas aeruginosa*

Then I check the references and found that the cited references are rather old, There is only one 2022 ref. and no 2023 ref.

More importantly, the authors have not really addressed my previous concern regarding the detail biochemical investigation of the YbfA as the target of Tyrosol. Which residues of YbfA interact with Tyrosol? Mutagenesis of these residues should be done and the mutated YbfA should not be able to bind with Tyrosol anymore.

For Fig 6B&7A, the authors need to provide full original EMSA photos of these figures. Now these two results look very similar and both contain three lanes, which is not convincing and even looks a bit fake. No probe name is shown in these figures. They also need positive controls and serial-diluted probe concentrations.

Nearly all RT-qPCR results are only in 1.2 fold change ranges, which is not of biological meaning according to my 20 years research experience when using RT-PCT analysis.

ITC titration was repeated only twice, which needs at least three repeats.

Apr. 4, 2024

I have revised the manuscript according to the comments made by the reviewer. The modified parts were highlighted in red color in the revised manuscript.

Modifications according to the comment of reviewer # 4:

1) While reviewing the revised version of this manuscript, I have these general opinion that this manuscript is lack of novelty. Tyrosol is not a very new compound and there are already papers reported it is a quorum sensing inhibitor and biofilm suppressor.

I) 2015 BioMed Research International paper :Effect of Tyrosol and Farnesol on Virulence and Antibiotic Resistance of Clinical Isolates of *Pseudomonas aeruginosa*”.

II) 2019 Bioorganic Chemistry paper :Tyrosol from marine Fungi, a novel Quorum sensing inhibitor against *Chromobacterium violaceum* and *Pseudomonas aeruginosa*.

Response) References 74 and 75; lines 647-648: The anti-QS and anti-biofilm activities of tyrosol were already mentioned in the first revision, with the two relevant papers cited.

We would like to emphasize a new function of tyrosol presented in our study. Although the title of the “2015 BioMed Research International” paper may suggest that tyrosol affect antibiotic resistance of clinical isolates of *P. aeruginosa*, this paper actually reported that tyrosol did not alter antibiotic sensitivity of *P. aeruginosa* clinical isolates in either disk diffusion or broth microdilution assays, which were conducted in planktonic, non-biofilm, cultures. In contrast, our study demonstrates that tyrosol significantly reduces antibiotic resistance of *P. aeruginosa* clinical isolates in biofilm cultures, especially through inhibition of anaerobic biofilm formation, by targeting YbfA. Thus, our findings represent the first report

of a new mechanism by which tyrosol enhances sensitivity to antibiotics in biofilm cells.

- 2) Then I check the references and found that the cited references are rather old, There is only one 2022 ref. and no 2023 ref.

Response) Some older references have been updated with more recent literatures as detailed below;

- (1) Reference 1: Flemming HC, Wingender J. “The biofilm matrix.” *Nat Rev Microbiol* **8**, 623-633 (2010) was updated to Flemming HC, *et al.* “The biofilm matrix: multitasking in a shared space.” *Nat Rev Microbiol* **21**, 70-86 (2023).
- (2) Reference 4: Ito A, Taniuchi A, May T, Kawata K, Okabe S. “Increased antibiotic resistance of Escherichia coli in mature biofilms.” *Appl Environ Microbiol* **75**, 4093-4100 (2009) was updated to Ciofu O, Moser C, Jensen PO, Hoiby N. “Tolerance and resistance of microbial biofilms.” *Nat Rev Microbiol* **20**, 621-635 (2022).
- (3) Reference 13: Driscoll JA, Brody SL, Kollef MH. “The epidemiology, pathogenesis and treatment of Pseudomonas aeruginosa infections.” *Drugs* **67**, 351-368 (2007) was updated to Qin S, *et al.* “Pseudomonas aeruginosa: pathogenesis, virulence factors, antibiotic resistance, interaction with host, technology advances and emerging therapeutics.” *Signal Transduct Target Ther* **7**, 199 (2022).
- (4) Reference 15: Smith K, Hunter IS. “Efficacy of common hospital biocides with biofilms of multi-drug resistant clinical isolates.” *J Med Microbiol* **57**, 966-973 (2008) was updated to Camara M, *et al.* “Economic significance of biofilms: a multidisciplinary and cross-sectoral challenge.” *NPJ Biofilms Microbiomes* **8**, 42 (2022).

- 3) More importantly, the authors have not really addressed my previous concern regarding the detail biochemical investigation of the YbfA as the target of Tyrosol. Which residues of YbfA interact with Tyrosol? Mutagenesis of these residues should be done and the mutated YbfA should not be able to bind with Tyrosol anymore.

Response) Firstly, I apologize for not addressing the question during the first revision due to a limited time.

Figs. 3C and S7; lines 275-304: Key amino acids in YbfA interacting with tyrosol were identified using a combination of protein structure prediction, molecular docking, and site-directed mutagenesis. Molecular docking of the predicted protein structure with tyrosol allowed for the prediction of potential amino acid residues interacting with tyrosol were predicted. Tyrosol was found to potentially form hydrogen bond and hydrophobic interactions with the amino acid residues. All six amino acid residues in the binding pocket were individually subjected to site-directed mutagenesis. Initially, in vivo binding activities of the resulting six YbfA variants to tyrosol were performed by investigating the effects of tyrosol on biofilm formation in strains overexpressing YbfA variants. These in vivo assays indicated that four YbfA variants (A19G, V20G, L43G and W47A) did not bind to tyrosol, while the other two (Y40A and H44A) did. Next, direct binding assays of the YbfA variants to tyrosol were conducted using ITC analysis. Consistent with the in vivo binding assay results, the ITC experiments confirmed that the four YbfA variants did not bind to tyrosol. Additionally, since these variants retained the biofilm-stimulating activity characteristic of the YbfA wild-type, the possibility that the site-directed mutations might induce significant alterations in the tertiary structure of YbfA, thereby abolishing tyrosol's binding affinity to YbfA variants, can be ruled out. Overall, this study demonstrates that tyrosol interacts with the residues Ala19, Val20, Leu43, and Trp 47 in YbfA.

- 4) For Fig 6B&7A, the authors need to provide full original EMSA photos of these figures. Now these two results look very similar and both contain three lanes,

which is not convincing and even looks a bit fake. No probe name is shown in these figures. They also need positive controls and serial-diluted probe concentrations.

Response)

Figs. 6B and 7A; lines 411-417 and 429-433: Since YbfA and YfeR are newly identified transcription factors in this study, appropriate positive controls for YbfA and YfeR are not available. Consequently, only negative controls were employed to demonstrate specific activities of the proteins. The 150-bp *fnr* promoter DNA (*Pfnr*) was used as a negative control for EMSA analysis of YbfA binding to the *yfeR* promoter. Conversely, the 150-bp *yfeR* promoter DNA (*PyfeR*) was used as a negative control for EMSA analysis of YfeR binding to the *fnr* promoter.

The concentrations of both proteins and unlabeled probes were varied. Varying concentrations of YbfA and YfeR (1, 3, and 10 nM) were tested. Serial dilutions of the unlabeled probes, at 50, 100, and 200-fold, were used for competition assays.

In Source data, the full original EMSA images of Figs. 6B and 7A, which represent typical results of chemiluminescent EMSA using biotin-labeled DNA, have been provided.

- 5) Nearly all RT-qPCR results are only in 1.2 fold change ranges, which is not of biological meaning according to my 20 years research experience when using RT-PCT analysis.

Response) As reviewer #4 mentioned, the 1.2 to 1.3-fold changes in *napA*, *norV*, *fnr*, and *yfeR* transcription observed by tyrosol treatment in RT-qPCR are modest; however, we believe that these changes are biologically significant for the following two main reasons.

First, the changes have high statistical significance. By combining data from all five independent experiments ($n = 15$) for *napA* and *fnr* RT-qPCR in Figures 1F, 2C, 4A11, 5Bii, and 7Bi, we observed extremely low p values, as shown in the figure below. The changes in *napA* and *fnr* transcription at 10 μ M tyrosol are 1.26-fold, with statistical significance values of $p = 7.3E-19$ and $1.1E-31$, respectively. More

importantly, these changes are dose-dependent, with p-values of 0.037 and 0.0023 for 1 vs 10 μM tyrosol, for *napA* and *fnr*, respectively. Furthermore, alterations in *napA* and *fnr* transcription are observed in a time dependent manner, beginning at 6 hr and peaking at 12 hr of biofilm cultivation (Fig. 8Bii).

Secondly, the 1.2 to 1.3-fold changes in *napA*, *norV*, *fnr*, and *yfeR* transcription suggest a role of *napA* in the tyrosol-mediated inhibition of biofilms. This hypothesis was confirmed through mutation, overexpression, and complementation experiments (Fig. 2). The mutation of *napA* abolished tyrosol's antibiofilm activity as well as its effects on NO production, c-di-GMP levels and related PDE activity. Complementation of *napA* in the *napA* mutant restored tyrosol's antibiofilm effects and related activity. Conversely, overexpression of *napA* also abolished tyrosol's antibiofilm activity and related activity. These findings underscore the essential role of *napA* in mediating tyrosol's antibiofilm activity. Furthermore, RT-qPCR analysis revealed that mutation of *napA* abolished its transcriptional response to tyrosol, which was restored upon complementation of *napA* in the *napA* mutant. Overexpression of *napA* similarly abolished its transcription response to tyrosol. Overall, these results indicate that tyrosol induces biofilm formation by stimulating *napA* transcription. Using the similar genetic approaches, the involvement of *fnr* and *yfeR* in tyrosol-mediated transcription was also demonstrated (Figs. 5 and 7).

Therefore, although the changes in *napA*, *fnr*, and *YfeR* transcription by tyrosol observed in RT-PCR are modest, genetic experiments have demonstrated that even slight increases in *napA*, *fnr*, and *YfeR* transcription are essential for initiating tyrosol's inhibitory effects on biofilm formation through mechanisms including NO production, PDE enhancement, and reduction in c-di-GMP level. Overall, we

believe that this study elucidates the biological significance of the modest changes in napA transcription.

The expression levels of proteins necessary to initiate an appropriate phenotype vary depending on the protein. It has been reported that low-expression proteins are energetically beneficial to cells, since protein expression is a metabolically costly process, and therefore more evolved compared to high-expression proteins (Genetics 202, 273, 2016; <https://doi.org/10.1534/genetics.115.180547>). The denitrification pathway is considered energetically less favorable due to its anaerobic process. It has also been reported that, consistent with the energetically unfavorable nitrate uptake, nitrate reductase NapA is synthesized only at very low nitrate levels and therefore the low-substrate nitrate reductase was suggested to have higher affinity for nitrate (J. Bacteriol.181, 5303, 1999; <https://doi.org/10.1128%2Fjb.181.17.5303-5308.1999>). In my humble opinion, given the energetically unfavorable nature of the denitrification pathway and the high affinity of napA for its substrate, it appears reasonable that proteins involved in the denitrification pathway, such as NapA, NorV, YfeR and Fnr, are expressed at low levels in this study.

In some literatures, fold changes of approximately 1.2 to 1.3 in RT-qPCR have also been reported as significant gene expression changes. Examples include Fig. 1 and Tables 1 and 2 in “J Bone Miner Res. 2016 Apr;31(4):839-51”(<https://doi.org/10.1002/jbmr.2752>); Tables 2 and 3 in “Diabetes 2004 Jun;53(6):1496-508” (<https://doi.org/10.2337/diabetes.53.6.1496>); Fig 1 in “Heliyon 9, e21658 (2023)” (<https://doi.org/10.1016/j.heliyon.2023.e21658>); Fig. 6 in “Rice (NY) 17, 16 (2024)” (<https://doi.org/10.1186/s12284-024-00694-z>); and Fig. 3 in “PLoS One 11, e0159028 (2016)” (<https://doi.org/10.1371/journal.pone.0159028>). Additionally, according to the “Diabetes 2004 Jun;53(6):1496-508” paper, a fold change of 1.3 has been used to identify significant gene expression change in some studies, including *J Biol Chem* 27: 49036–49046, 2002; *J Biol Chem* 278:15633–15640, 2003; and *Diabetes* 50:2268–2278, 2001.

6) ITC titration was repeated only twice, which needs at least three repeats.

Response) In the Source Data, Fig. 3Civ: ITC titration for all tested proteins was performed three times.

If there is any question about the manuscripts, please let me know. Your consideration will be very much appreciated.

REVIEWERS' COMMENTS

Reviewer #4 (Remarks to the Author):

The authors have performed the key biochemistry experiments to strengthen their findings. I am satisfied about this version of manuscript.